



# Decadal trajectories of nitrate input and output in three nested catchments along a land use gradient

Sophie Ehrhardt[1], Rohini Kumar[2], Jan H. Fleckenstein[3], Sabine Attinger[2], Andreas Musolff [1]

[1]Department of Hydrogeology, Helmholtz-Centre for Environmental Research, Leipzig, 04318, Germany
[2]Department Computational Hydrosystems, Helmholtz-Centre for Environmental, Leipzig, 04318, Germany
[3] Bayreuth Center of Ecology and Environmental Research, University of Bayreuth, Bayreuth, 95440, Germany

*Correspondence to*: Andreas Musolff (andreas.musolff@ufz.de)

**Abstract.** Increased anthropogenic inputs of nitrogen (N) to the biosphere during the last decades have resulted in increased
groundwater and surface water concentrations of N (primarily as nitrate) posing a global problem. Although measures have
been implemented to reduce N-inputs especially from agricultural sources, they have not always led to decreasing riverine
nitrate concentrations and loads. The limited response to the measures can either be caused by the accumulation of slowly
mineralized organic N in the soils acting as a biogeochemical legacy or by long travel times (TTs) of inorganic N to the
streams forming a hydrological legacy. Both types of legacy are hard to distinguish from the TTs and N budgets alone. Here
we jointly analyze atmospheric and agricultural N inputs with long-term observations of nitrate concentrations and discharge
in a mesoscale catchment in Central Germany. For three nested sub-catchments with increasing agricultural land use, we
assess the catchment scale N budget, the effective TT of N. In combination with long-term trajectories of C-Q relationships
we finally evaluate the potential for and the characteristics of an N-legacy. We show that in the 42-year-long observation
period, the catchment received an N-input of 42 758 t, of which 97 % derived from agricultural sources. The riverine N-
export sums up to 6 592 t indicating that the catchment retained 85 % of the N-input. Removal of N by denitrification could
not fully explain this imbalance. Log-normal travel time distributions (TTD) for N that link the input history to the riverine
export differed seasonally, with modes spanning 8–17 years. Under low-flow conditions, TTs were found to be
systematically longer than during high discharges. Systematic shifts in the C-Q relationships could be attributed to
significant changes in N-inputs resulting from agricultural intensification and the break-down of the East German agriculture
after 1989 and to the longer travel times of nitrate during low flows compared to high flows. A chemostatic export regime of
nitrate was only found after several years of stabilized N-inputs. We explain these observations by the vertical migration of
the N-input and the seasonally changing contribution of subsurface flow paths with differing ages and thus differing N-loads.
The changes in C-Q relationships suggest a dominance of hydrological N-legacy rather than a biogeochemical N-fixation in
the soils, which should result in a stronger and even increasing dampening of riverine N-concentrations after sustained high
N-inputs. Despite the strong N-legacy, a chemostatic nitrate export regime is not necessarily a persistent endpoint of intense
agricultural land use, but rather depends on a steady replenishment of the mass of N propagating through the catchments
subsurface. The input-output imbalance, the long time-lags and the lack of significant denitrification in the catchment let us





conclude that catchment management needs to address both, a longer-term reduction of N-inputs and shorter-term mitigation of today's high N-loads.

# 1 Introduction

In terrestrial, freshwater and marine ecosystems nitrogen (N) species are essential and often limiting nutrients (Webster et al., 2003). Changes in strength of their different sources like atmospheric deposition, wastewater inputs as well as agricultural activities caused major changes in the terrestrial nitrogen cycle (Webster et al., 2003). Especially two major innovations from the industrial age accelerated anthropogenic inputs of reactive nitrogen species into the environment: artificial nitrogen fixation and the internal combustion engine (Elser, 2011). The anthropogenically released amount of reactive nitrogen that enters the element's biospheric cycle has been doubled in comparison to the preindustrial era (Smil et al., 1999). However, the different input sources of nitrogen show diverging trends over time and space. While the atmospheric emissions of nitrogen oxides and ammonia declined in Europe since the 1980s (EEA, 2014), the agricultural nitrogen input (N-input) through fertilizers is still high (Federal Ministry for the Environment and Federal Ministry of Food, 2012). Consequently, in the cultural landscape of Western countries, most of the nitrogen emissions in surface and groundwater bodies stem from diffuse agricultural sources (Bouraoui and Grizzetti, 2011; Dupas et al., 2013).

The widespread consequences of these excessive N-inputs are significantly elevated concentrations of dissolved inorganic nitrogen (DIN) in groundwater and connected surface waters (Altman and Parizek, 1995; Sebilo et al., 2013; Wassenaar, 1995) as well as the associated increases in riverine DIN fluxes (Dupas et al., 2016) causing the ecological degradation of freshwater and marine systems. This degradation is caused by the ability of nitrogen species to increase primary production and to change food web structures (Howarth et al., 1996; Turner & Rabalais, 1991). Especially the coastal marine environments, where nitrate ($NO_3$) is typically the limiting nutrient, are affected by these eutrophication problems (Decrem et al., 2007; Prasuhn and Sieber, 2005).

To cope with this problem, international, national and federal regulations have been implemented aiming at a reduction of N-inputs into the terrestrial system and its transfer to the aquatic system. In the European Union, guidelines are provided to its member states for national programs of measures and evaluation protocols through the Nitrate Directive (CEC, 1991) and the Water Framework Directive (CEC, 2000).

The evaluation of the measures showed that policy-makers struggle to set appropriate goals for water quality improvement in human-impacted watersheds, as the reduced N-inputs mainly in agricultural land use, are often not immediately resulting in decreasing riverine nitrate concentrations (Bouraoui and Grizzetti, 2011) and fluxes. Also in Germany considerable progress has been achieved towards the improvement of water quality, but the diffuse water pollution from agricultural sources continues to be of concern (Wendland et al., 2005). This limited response to mitigation measures can partly be explained by nutrient legacy effects stemming from an accumulation of excessive fertilizer inputs over decades creating a time lag between the implementation of measures and water quality improvement (van Meter & Basu, 2015). Furthermore, the multi-





year transfer time of nitrate through the unsaturated and saturated zones of the catchment itself causes large time lags (Howden et al., 2010; Melland et al., 2012) that can mask the riverine response to measures. We therefore need a profound understanding of the processes and controls of time lags of N from the source to groundwater and surface water.

Calculating the travel time of water or solutes through the landscape is essential for predicting the retention, mobility and
fate of solutes, nutrients and contaminants at catchment-scale (Jasechko et al., 2016). Time series of concentrations and loads that cover both, input to the geosphere and the subsequent riverine export, can be used not only to determine travel times (TTs; van Meter & Basu, 2017) but also to quantify mass losses in the export or the catchment's retention capacity, respectively (Dupas et al., 2015).

Data driven or simplified mechanistic approaches have often been used to derive stationary and seasonally variable TT
distributions using in- and output signals of conservative tracers or isotopes (Jasechko et al., 2016; Heidbüchel et al., 2012) or chloride concentrations (Kirchner et al., 2000; Bennettin et al., 2015). However, recently van Meter & Basu (2017) estimated the solute TTs for nitrogen transport at several stations across a catchment located in Southern Ontario, Canada, showing decadal time-lags between input and riverine exports. Moreover, systematic seasonal variations in nitrate concentrations have been found, which were explained by seasonal shifts in the nitrogen delivery pathways and connected
time lags (van Meter & Basu, 2017). Despite the determination of these seasonal concentration and age dynamics, there is generally a lack of studies focussing on their long-term trajectory under conditions of changing N-inputs. Seasonally differing time shifts, resulting in changing intra-annual concentration variations are of importance for aquatic ecosystem health and functionality. Seasonal concentration changes can also be directly connected to changing concentration–discharge (C–Q) relationships – a tool for classifying solute responses to changing discharge conditions and for characterizing and
understanding anthropogenic impacts on solute input, transport and fate (Jawitz & Mitchell, 2011; Musolff et al. 2015).

C–Q relationships can be on the one hand classified in terms of their pattern characterized by the slope b of the lnC–lnQ regression (Godsey et al., 2009): with enrichment (b>0), dilution (b<0) or constant (b≈0) patterns (Musolff et al., 2017). On the other hand, C–Q relationships can be classified according to the ratio between the coefficients of variation of concentration ($CV_C$) and discharge ($CV_Q$; Thompson et al., 2011). This export regime can be either chemodynamic
($CV_C/CV_Q > 0.5$) or chemostatic, where the variance of the solute load is more strongly dominated by the variance in discharge than the variance in concentration (Musolff et al., 2017). Both, patterns and regimes are dominantly shaped by the spatial distribution of solute sources (Seibert et al., 2009; Basu et al., 2010; Thompson et al., 2011; Musolff et al., 2017). High source heterogeneity and consequently high concentration variability in the discharge is thought to be characteristic for nutrients under pristine conditions (Musolff et al., 2017, Basu et al., 2010). It was shown that catchments under intensive
agricultural use evolve from chemodynamic to more chemostatic behavior regarding nitrate export (Thompson et al., 2011; Dupas et al., 2016). Several decades of human N-inputs seem to dampen the discharge-dependent concentration variability, resulting in chemostatic behavior where concentrations are largely independent of discharge variations (Dupas et al., 2016). Also Thompson et al. (2011) stated observational and model-based evidence of an increasing chemostatic response of nitrate with increasing agricultural intensity. It has been argued that this shift in the export regimes is caused by a long-term





homogenisation of the nitrate sources in space and/ or in depth within soils and aquifers (Dupas et al., 2016; Musolff et al., 2017). Long-term N inputs lead to a loading of all flow paths in the catchment with mobile fractions of N and by that to the formation of a hydrological N-legacy (van Meter et al., 2015) and chemostatic riverine N exports. On the other hand, excessive fertilizer input is linked to the build-up of legacy nitrogen stores in the catchment, changing the export regime

from a supply- to a transport-limited chemostatic one (Basu et al., 2010). This legacy is manifested as a biogeochemical legacy in form of increased, less mobile, organic N content within the soil (Worral et al., 2015; van Meter et al., 2015; van Meter et al., 2017a). This type of legacy buffers biogeochemical variations, so that management measures can only show effect if the build-up source gets substantially depleted (Basu et al., 2010). Depending on the catchment configuration, both forms of legacy can exist with different shares of the total nitrogen stored in a catchment (van Meter et al., 2017a). However,

biogeochemical legacy is hard to distinguish from hydrological legacy when looking at time lags between input and output or at catchment scale N budgets (van Meter et al., 2015). Here, the framework of C-Q relationships defined in Jawitz & Mitchell (2011), Musolff et al. (2015) and Musolff et al. (2017) can help to better disentangle N-legacy types: In case of a hydrological legacy strong changes of fertilizer inputs (such as increasing inputs in the initial phase of intensification and decreasing inputs as a consequence of measures) will temporarily increase spatial concentration heterogeneity (comparing

young and old water fractions in the catchment storage) and therefore also shift the export regime to more chemodynamic conditions. On the other hand, a dominant biogeochemical legacy will lead to a sustained concentration homogeneity in the N source zone in the soils and to an insensitivity of the riverine N export regime to fast changes in inputs.

Common approaches to quantify catchment scale nitrogen budgets (N-budgets) and to characterize legacy or to derive TTs are either based on data-driven top down approaches (Worral et al., 2015; Dupas et al., 2016) or on forward modeling (van

Meter et al., 2015; van Meter et al., 2017a). So far, the data-driven studies focused either solely on N-budgeting and legacy estimation or on TTs. Here we aim at a joint data-driven assessment of catchment scale N-budget, the potential and characteristics of a nitrogen legacy (N-legacy) and on the effective TTs of the riverine exported nitrogen. More specifically, we estimate N-budgets and effective nitrogen TT of a catchment from the same data base. Furthermore, we utilize the trajectory of agricultural catchments in terms of C–Q relationships, their changes over longer time scales and their potential

evolution to a chemostatic export regime to better disentangle legacy types. With these objectives, we aim to provide a better understanding of nitrogen retention capacity and transport mechanisms as a basis for a discussion of more effective catchment management. This study will address the following research questions:

1.  How high is the retention potential for N of the studied mesoscale catchment and what are the consequences in terms of a build-up of an N-legacy?

2.  What are the characteristics of the TT distribution for nitrogen that links change in the diffuse anthropogenic N-inputs to the geosphere and their observable effect in riverine nitrate concentrations?

3.  What are the characteristics of a long-term trajectory of C–Q relationships? Is there an evolution to a chemostatic export regime linked to evolving biogeochemical or hydrological N-legacies?





To answer these questions, we used time series of water quality data over four decades, available from a mesoscale German catchment, as well as estimated N-input to the geosphere. We link in- and output on annual and intra-annual time scales by the use of effective TT distributions. This input-output assessment uses time series of the Holtemme catchment (282 km²) with its three nested subcatchments along a land use gradient from pristine mountainous headwaters to a lower basin with

intensive agriculture and associated increases of fertilizer applications. This catchment with its pronounced increase in anthropogenic impacts from up- to downstream is quite typical for many mesoscale catchments in Germany and elsewhere. Moreover, this catchment offers a unique chance to utilize the strong changes in fertilizer usage in East-Germany before and after reunification. Thereby we anticipate that our improved understanding gained through this study in these catchment settings is transferable to other regions. In comparison to spatially and temporally integrated water quality signals stemming

solely from the catchment outlet, the higher spatial resolution with three stations and the unique length of the monitoring period allow for a more detailed information about the fate of nitrogen in the catchment and consequently favors a more effective river management.

## 2 Material and Methods

### 2.1 Study area

The Holtemme catchment (282 km²) is a subcatchment of the Bode River basin, which is part of the TERENO Harz/Central German Lowland Observatory (Fig. 1). The catchment was selected as part of the TERENO (**TER**restrial **EN**vironmental **O**bservatories) project because of its strong gradients in topography, climate, geology, soils, water quality, land use and level of urbanization (Wollschläger et al., 2017). Furthermore, the region is ranked as highly vulnerable to climate change (Schröter et al., 2005), representative for other German and central European regions showing similar vulnerability

(Zacharias et al., 2011). The observatory is one of the meteorologically and hydrologically best-instrumented catchments in Germany (Zacharias et al., 2011; Wollschläger et al., 2017), and provides long-term data for many environmental variables including water quantity (e.g. precipitation, discharge) and water quality at various locations. The Holtemme catchment has its spring at 862 m a.s.l. in the Harz Mountains and extends to the Northeast to the Central German Lowlands with an outlet at 85 m a.s.l.. The long-term annual mean precipitation (1951–2015) shows a decrease from colder and humid climate in the

Harz Mountains (1262 mm) down to the warmer and dryer climate of the Central German Lowlands (614 mm; Rauthe et al., 2013; Frick et al., 2014). Discharge time series, provided by the State Office of Flood Protection and Water Management (LHW) Saxony-Anhalt show a mean annual discharge at the outlet in Nienhagen of 1.5 m³ s$^{-1}$ (1976–2016) referring to 172 mm a$^{-1}$.

The geology is dominated by late Paleozoic rocks in the mountainous upstream part that are largely covered by Mesozoic

rocks as well as Tertiary and Quaternary sediments in the lowlands (Frühauf & Schwab, 2008; Schuberth, 2008). Land use of the catchment changes from forests in the pristine, mountainous headwaters to intensive agricultural use in the downstream lowlands (EEA, 2012). According to Corine Land Cover (CLC) from different years (1990, 2000, 2006, 2012), the land use



change over the investigated period is negligible. Overall 60 % of the catchment is used by agriculture, while 30 % is covered by forest (EEA, 2012). Urban land use occupies 8 % of the total catchment area (EEA, 2012) with two major towns (Wernigerode, Halberstadt) and several smaller villages. Two wastewater treatment plants (WWTPs) discharge into the river. The town of Wernigerode had its WWTP within its city boundaries until 1995, when a new WWTP was put into

5    operation about 9.1 km downstream in a smaller village, called Silstedt, replacing the old WWTP. The WWTP in Halberstadt was not relocated but renovated in 2000. Nowadays, the total nitrogen load (TNb) in cleaned water is approximately 67.95 kg d$^{-1}$ (WWTP Silstedt: $NO_3$-N load 55 kg d$^{-1}$) and 35.09 kg d$^{-1}$ (WWTP Halberstadt: $NO_3$-N load 6.7 kg d$^{-1}$; mean daily loads 2014; Müller et al., 2018). Despite this N-input, major nitrate contribution in recent years was related to agricultural land use (Müller et al., 2018).

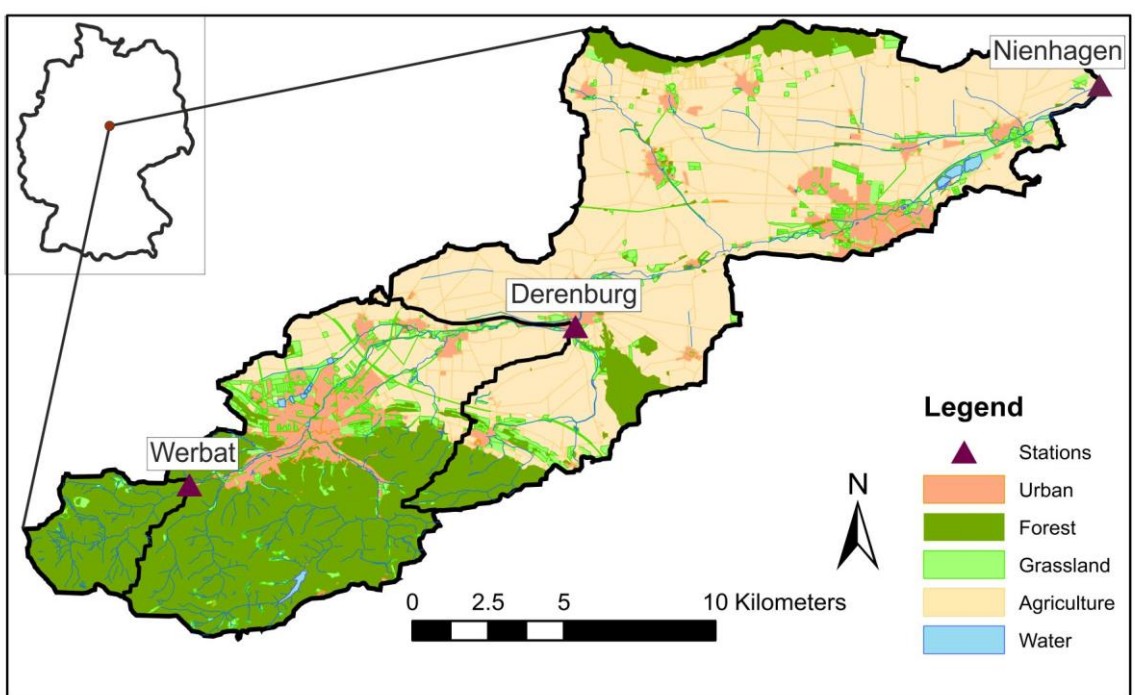

**Figure 1: Map of the Holtemme catchment with the selected sampling locations.**

The Holtemme River has a length of 47 km. Along the river, the LHW Saxony-Anhalt maintains long-term monitoring stations to provide daily mean discharge and biweekly to monthly water quality measurements. Three of the water quality

15    stations along the river were selected to represent the characteristic land use and topographic gradient in the catchment. From up- to downstream the stations are named Werbat, Derenburg and Nienhagen (Fig. 1Figure 1) in the following referred to as Upstream, Midstream and Downstream. The pristine headwaters upstream represent the smallest (6 % of total catchment




area) and at the same time steepest of the three selected subcatchments as it has about a three times higher mean topographic slope than the downstream parts (DGM25; Table1). According to the latest CLC from 2012, the land use is characterized by forest only. The larger midstream subcatchment that represents one third of the total area is still dominated by forests, but with growing anthropogenic impact due to increasing agricultural land use and the town of Wernigerode. In this

5  subcatchment more than half of the agricultural land is artificially drained (LHW, 2011; Table 1; S1.1). The largest subcatchment (61 %) constituting the downstream part, is located in the lowlands. This Central German lowlands are predominantly covered by Chernozems (Schuberth, 2008), which are the most fertile soils within Germany (Schmidt, 1995). Hence, the agricultural land use in this subcatchment is the highest in comparison to the two upstream subcatchments and makes up 81 % (EEA, 2012). Also the second town, Halberstadt, increases the anthropogenic impact to the Holtemme River.

10  The time series of the three stations along the Holtemme River cover roughly the last four decades (1970–2016) and represent the N-output of the input-output assessment.

**Table 1: Subcatchment information. n – number of observations.**

|  | Upstream | Midstream | Downstream |
|---|---|---|---|
| **n Q** | 16 132 | - | 12 114 |
| **n nitrate-N (NO$_3$-N)** | 646 | 631 | 770 |
| **Period of NO$_3$-N time series** | 1972–2014 | 1970–2011 | 1976–2016 |
|  |  |  |  |
| **Subcatchment area (km²)** | 15.06 | 88.50 | 165.22 |
| **Cumulative catchment area (km²)** | 15.06 | 103.60 | 268.80 |
| **Stream length (km)** | 1.5 | 19.3 | 24.4 |
| **Mean topographic slope** | 9.82 | 7.52 | 2.55 |
| **Mean topo. slope: non-forested area** | - | 3.2 | 1.9 |
|  |  |  |  |
| **Land use (EEA, 2012)** |  |  |  |
| **Forest land use (%)** | 100 | 56 | 11 |
| **Urban land use (%)** | - | 17 | 8 |
| **Agricultural land use (%)** | - | 27 | 81 |
| **Fraction of agricultural area artificially drained (%)** | - | 59.1 | 20.5 |



## 2.2 Nitrogen input

The main N-sources had to be quantified over time for the data-based input-output assessment. A recent investigation in the study catchment by Müller et al. (2018) showed that the major nitrate contribution stems from agricultural land use and the associated application of fertilizers. The quantification of this contribution is the N-surplus (also referred to as agricultural

surplus) that reflects N-inputs that are in excess of crop and forage needs.

The annual agricultural N-input for the Holtemme catchment was calculated using two different data sets of agricultural N-surplus across Germany provided by the University of Gießen (Bach & Frede, 1998; Bach et al., 2011). Surplus data [kg N ha$^{-1}$ a$^{-1}$] were available on the federal state level for 1950–2015 and on the county level for 1995–2015. We used the data from the overlapping time period (1995–2015) to downscale the state level data (state: Saxony-Anhalt) to the county level

(county: Harzkreis). Both data sets show high correspondence with a correlation ($R^2$) of 0.85, but they slightly differ in their absolute values (by 6 % of the mean annual values). The mean offset of 3.85 kg N ha$^{-1}$ a$^{-1}$ was subtracted from the federal state level data to yield the surplus in the county before 1995.

Both of the above datasets account for the atmospheric deposition, but only on agricultural areas. For other non-agricultural areas (forest and urban landscapes), the N-source stemming from atmospheric deposition was quantified based on datasets

from the Meteorological Synthesizing Centre - West (MSC-W) of the European Monitoring and Evaluation Programme (EMEP). The underlying dataset consists of gridded fields of EU-wide wet and dry atmospheric N-depositions from a chemical transport model that assimilates different sources of EU-wide observational datasets on different atmospheric chemicals (e.g. Bartnicki & Benedictow, 2017; Bartnicki & Fagerli, 2006). This dataset is available at annual time-steps since 1995, and at every 5 a between 1980 and 1995. Data between the 5-a-dataset were linearly interpolated to obtain

annual estimates of N-deposition between 1980 and 1995. For years prior to 1980, we made use of global gridded estimates of atmospheric N-deposition from the three-dimensional chemistry-transport model (TM3) for the year 1860 (Dentener, 2006; Galloway et al., 2004). In absence of any other information, we performed a linear interpolation of the N-deposition estimates between 1860 and 1980.

To quantify the net N-fluxes to the soil via atmospheric deposition, the terrestrial biological N-fixation had to be subtracted

for different non-agricultural land use types. Based on a global inventory of terrestrial biological N-fixation in natural ecosystems, Cleveland et al. (1999) estimated the mean uptake for temperate (mixed, coniferous or deciduous) forests and (tall/medium or short) grassland as 16.04 kg N ha$^{-1}$ a$^{-1}$, and 2.7 kg N ha$^{-1}$ a$^{-1}$, respectively. The remaining atmospheric deposition after biological fixation, calculated for the different land use amounts in each subcatchment, was added to the agricultural N-surplus to achieve the total N-input per area and subcatchment. In contrast to the widely applied term net

anthropogenic nitrogen input (NANI), we do not account for wastewater fluxes at this point but rather focus on the diffuse atmospheric deposition, biological N-fixation and agricultural input.





## 2.3 Nitrogen output

### 2.3.1 Discharge and water quality time series

The data for water quality (biweekly to monthly) and discharge (daily) from 1970 to 2016 were provided by the LHW, Saxony-Anhalt.

5    The biweekly to monthly sampling was done at gauging stations defining the three subcatchments (NO$_3$-N: Fig. 2; NH$_4$-N: S1.2.1; NO$_2$-N: S1.2.2). The data sets cover a wide range of instream chemical constituents including major ions, alkalinity, nutrients and in-situ parameters. As this study only focuses on N-species, we restricted the selection of parameters to nitrate (NO$_3$), nitrite (NO$_2$) and ammonium (NH$_4$).





**Figure 2: NO$_3$-N concentration time series: Upstream (a), Midstream (b) and Downstream (c).**

Discharge time-series at daily time scales were measured at two of the water quality stations (Upstream, Downstream; Fig. 3). Continuous daily discharge series are required to calculate flow-normalized concentrations. To derive the discharge data

5   for the midstream station and to fill measurement gaps at the other stations (2 % Upstream, 3 % Downstream), we used simulations from a grid-based distributed mesoscale hydrological model, called mHM (Samaniego et al., 2010; Kumar et al., 2013). Daily mean discharge was simulated for the same time frame as the available measured data. We used a model set-up similar to Müller et al. (2016) with robust results capturing the observed variability of discharge in the studied, near-by catchments. We note that the discharge time series is used as weighting factors in the later analysis of flow-normalized

10  concentrations. Consequently it is more important to capture the temporal dynamics than the absolute values. Nonetheless, we performed a simple bias correction method by applying the regression equation of simulated and measured values to reduce the simulated bias of discharge. After this revision, the simulated discharges could be used to fill the gaps of measured data. The midstream station (Derenburg) for water quality data is 5.6 km upstream of the next gauging station. Therefore, the nearest station (Mahndorf) with simulated and measured discharge data was used to derive a bias correction

15  equation that was subsequently applied to correct the simulated discharge data Midstream assuming the same bias between model and measurement in both stations.



**Figure 3: Discharge time series: Upstream (a), Midstream (b) and Downstream (c).**

**2.3.2 Weighted regression on time, discharge and season (WRTDS) and waste water correction**

5    The software package "Exploration and Graphics for RivEr Trends" (EGRET) in the R software environment by Hirsch and De Cicco (2010) was used to derive flow-normalized concentrations of NO$_3$-N. This tool enables an analysis, based on the long-term changes in water quality and streamflow, using the water quality method "Weighted Regressions on Time, Discharge, and Season" (WRTDS; Hirsch & De Cicco, 2010). The WRTDS method allows to increase the temporal resolution of concentration measurements to the daily scale using a flexible statistical representation for every day of the





discharge record (Hirsch & De Cicco, 2010). Both data sets on a daily resolution (discharge, concentration) were subsequently used to calculate two different time series: 1. Daily, flow-normalized concentrations, and 2. Daily, flow-normalized fluxes. Flow-normalization uses the probability distribution of discharge of the specific day of the year from the entire discharge time series. More specifically, the flow-normalized concentration is the average of the same regression

model for a specific day applied to all measured discharge values of the corresponding day of the year. While the non-flow-normalized concentrations are strongly dependent on the discharge, the flow-normalized estimations provide a more unbiased, robust estimate of the concentrations with a focus on changes in concentration and fluxes independent of inter-annual discharge variability (Hirsch & De Cicco, 2010).

The study of Müller et al. (2018) indicated a dominance of nitrogen from diffuse sources in the Holtemme catchment, but

also stressed an impact of wastewater-borne $NO_3$ during low flow periods. Since we aim at balancing and comparing N-input and outputs from diffuse sources only, the provided annual flux of total N from the two WWTPs was therefore used to correct flow-normalized fluxes and concentrations derived from the WRTDS assessment. We argue that the annual wastewater N-flux is robust to correct the flow-normalized concentrations but does not allow for the correction of actually measured concentration data at a specific day. Both treatment plants provided snapshot samples of both, $NO_3$-N and total N-

fluxes, to derive the fraction of N that is discharged as $NO_3$-N. For the WWTP Halberstadt (384 measurements between January 2014 to July 2016) this fraction is 19 %, for Silstedt (eight measurements from February 2007 to December 2017) 81 %. We argue that the fraction of N leaving as $NH_4$, $NO_2$ and $N_{org}$ does not interfere with the $NO_3$-N flux in the river due to the limited length and therefore nitrification potential of the Holtemme River impacted by wastewater (see also S 1.2.3). We related the wastewater-borne $NO_3$-N flux to the flow-normalized daily flux of $NO_3$-N from the WRTDS method to get a

daily fraction of wastewater $NO_3$-N in the river that we used to correct the flow-normalized concentrations. Note that this correction was applied to the midstream station from 1996 on when the Silstedt treatment plant was taken to operation. In the downstream station, we additionally applied the correction from the Halberstadt treatment plant, renovated in the year 2000. Before that, we assume that waste water-borne N dominantly leaves the treatment plants as $NH_4$-N (see also S1.2.1).

Based on the daily resolved flow-normalized and wastewater-corrected concentration and flux data, descriptive statistical

metrics were calculated on an annual time scale. Seasonal statistics of each year were also calculated for winter (December, January, February), spring (March, April, May), summer (June, July, August) and fall (September, October, November). Note that winter statistics incorporate December values from the calendar year before.

Following Musolff et al. (2015, 2017), the ratio of $CV_C/CV_Q$ and the slope (b) of the linear relationship between ln C and ln Q were used to characterize the export pattern and the export regimes of $NO_3$-N along the three study catchments.

## 2.4 Input-output assessment

The stream concentration of a given solute, e.g. as shown by Kirchner et al. (2000), is assumed at any time as the convolution of the travel time distribution (TTD) and the rainfall concentration throughout the past. This study applies the



same principle for the N-input as incoming time series that, when convolved with the TTD, yields the stream concentration time series. We selected a log-normal distribution function (with two parameters; μ and σ) as a convolution transfer function; based on a recent study by Musolff et al. (2017) who successfully applied this form of a transfer function to represent TTs. The two free parameters were obtained through optimization based on minimizing the sum of squared errors between observed and simulated N-exports. The form of selected transfer function is in line with Kirchner et al. (2000) stating that exponential TTDs are unlikely at catchment scale but rather a skewed, long tailed distribution. Note that we used the log-normal distribution as a transfer function between the temporal patterns of input (N-load per area) and flow-normalized concentrations on an annual base only and not as a flux-conservative transfer.

## 3 Results

### 3.1 Input assessment

In the period from 1950 to 2015, the Holtemme catchment received a cumulative diffuse N-input of 62 335 t. From this sum, the major part (97 %) can be associated with agriculture. Within the period where water quality data were available, the total sum is 51 091 t (1970–2015), as well with 97 % agricultural contribution. The N-input showed a remarkable temporal variability (Fig. 7; purple line). From 1950 to 1976, the input was characterized by a strong increase (slope of linear increase = 4.2) with a maximum annual, agricultural input of 132.05 kg N ha$^{-1}$ a$^{-1}$ (1976), which is twenty times the agricultural input from 1950. After more than 10 a of high but more stable inputs, the N-surplus dropped dramatically with the peaceful reunification of Germany and the collapse of the established agricultural structures (1989/1990; Gross, 1996). In the time period afterwards (1990–1995), the N-surplus was only one-sixth (20 kg N ha$^{-1}$ a$^{-1}$) of the previous input. After another 8 a of increased agricultural inputs (1995–2003) of around 50 kg N ha$^{-1}$ a$^{-1}$, the input slowly decreased with a mean slope of -1.3 kg N ha$^{-1}$ a$^{-1}$ per year, but showed distinctive changes in the input between the years.

The input into the forested catchment upstream (only atmospheric deposition) peaked 1980 and decreased afterwards. All of the annual inputs were below 12 kg N ha$^{-1}$ a$^{-1}$, which is less than one-fifth of the mean agricultural input (60 kg N ha$^{-1}$ a$^{-1}$). Hence, the input to the upstream area was only minor in comparison to the ones further downstream that are dominated by agriculture.

### 3.2 Output assessment

#### 3.2.1 Discharge time series and WRTDS results on decadal statistics

Discharge was characterized by a strong seasonality throughout the entire data record, which divided the year into a High-Flow-Season (HFS) during winter and spring accounting for two-thirds of the annual discharge and a Low-Flow-Season (LFS) during summer and fall. The upstream subcatchment contributed 21 % of the median discharge measured at the downstream station (Table 2). The midstream station, representing the cumulated discharge signal from the up- and





midstream subcatchments, accounted for 82 % of the median annual discharge at the outlet. Although the upstream subcatchment had the highest specific discharge, the major fraction of total discharge (61%) was generated in the midstream subcatchment. Also the seasonality in discharge was dominated by this major midstream contribution, especially during high flow conditions. Vice versa, especially during HFSs, the median downstream contribution was <10 %, while during low flow

periods, the downstream contribution accounted for up to 33 % (summer).

**Table 2: Descriptive statistics on discharge at the three observation points. LFS – low flow season (June–November), HFS – high flow season (December–May).**

|                                          | Upstream | Midstream | Downstream |
|------------------------------------------|----------|-----------|------------|
| **Median discharge (m³ s⁻¹)**            | 0.23     | 0.9       | 1.1        |
| **Mean specific discharge (mm a⁻¹)**     | 768      | 411       | 178        |
| **LFS contribution (%)**                 | 17       | 70        | 100        |
| **HFS contribution (%)**                 | 21       | 90        | 100        |

The flow-normalized $NO_3$-N concentrations in each subcatchment showed strong differences in their general level and temporal patterns over the four decades (Fig. 4a, see also Fig.2). The lowest decadal concentration changes and the earliest decrease in concentrations were found in the pristine catchment. Median upstream concentrations were highest in the 80s (1987), with a reduction of the concentrations to about one half in the decades afterwards. Over the entire period, the median upstream concentrations were smaller than 1 mg $L^{-1}$, so that the described changes are small compared to the $NO_3$**-N**

dynamics of the more downstream stations. High changes over time were observed in the two downstream stations with a tripling of concentrations between the 70s and 90s, when maximum concentrations were reached. While median concentrations downstream decreased slightly after this peak (1995/1996), the ones midstream (peak: 1998) stayed constantly high. At the end of the observation period, at the outlet (Downstream), the median annual concentrations did not decrease below 3 mg $L^{-1}$ $NO_3$-N, a level that was exceeded after the 70s. The differences in $NO_3$-N concentrations between

the pristine upstream and the downstream station evolved from an increase by a factor of 3 in the 70s to a factor of 7 after the 80s.



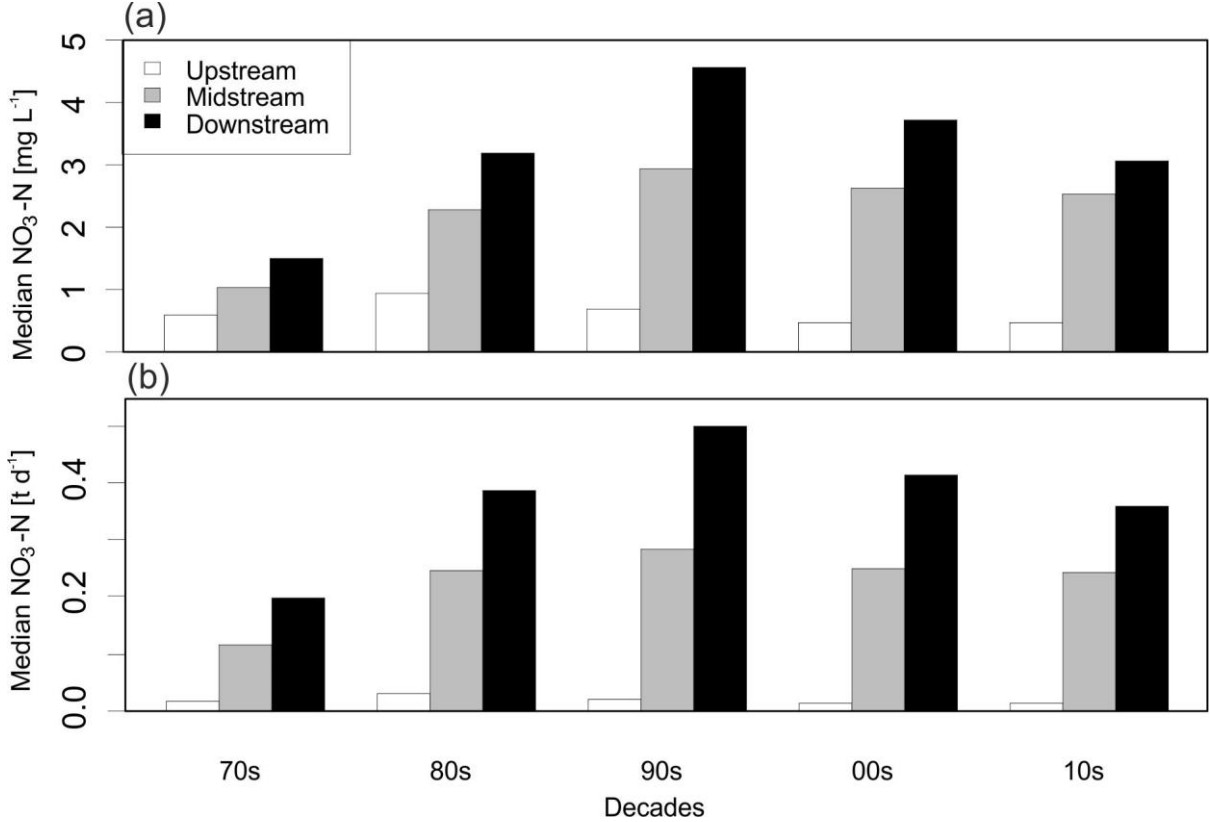

**Figure 4: Flow-normalized median NO$_3$-N concentration (a) and NO$_3$-N loads (b) for each decade of the time series and the three stations.**

Calculated loads (Fig. 4b) also showed a drastic change between the beginning and the end of the time series. The daily upstream load contribution was below 10 % of the total annual export at the downstream station in all decades and decreased from 9 % (70s) to 4 % (10s). The median daily load between 70s and 90s tripled midstream (0.1 t d$^{-1}$ to 0.3 t d$^{-1}$) and more than doubled downstream (0.2 t d$^{-1}$ to 0.5 t d$^{-1}$). In the 90s, the Holtemme River exported on average more than 0.5 t d$^{-1}$ of NO$_3$-N, which, related to the agricultural area in the catchment, translates into more than 3.1 kg N d$^{-1}$ km$^{-2}$ (max. 13.4 kg N ha$^{-1}$ a$^{-1}$, 1995).

## 3.3 Input-Output-balance: N-budget

The estimated N-inputs were associated with the exported loads of the subcatchment besides the statistical evaluation of the time series. This connection on the one hand allowed for an estimation of the catchment's retention potential with a discussion on potentially accumulated biogeochemical and hydrological legacy, and on the other hand it enabled us to predict future exportable loads.





**Table 3: Overview about derived retention potentials derived for the midstream and downstream subcatchment.**

|  | Midstream | Downstream |
|---|---|---|
| Retention cumulative (%) | 54 (Up + Midstream) | 85 (Up + Mid + Downstream) |
| Retention subcatchment (%) | 48 | 94 |
| Retention/Year (N kg a$^{-1}$) | 86 282 | 910 349 |
| Retention/Area (N kg a$^{-1}$ ha$^2$) | 9.75 | 55.10 |

The load stemming from the most upstream, pristine catchment accounted for <10 % of the exported load at the outlet. To focus on the anthropogenic impacts on catchments, the data from the upstream station are not discussed on its own in the

5  following. At the midstream station, a total sum of input of 7 653 t resulted in 4 109 t of exported $NO_3$-N for the overlapping time period of in- and output (1970–2011). Hence, 46 % (Table 3) of the applied N was exported in this period by the Holtemme River. With the assumption that 97 % of the diffuse input resulted from agriculture, the catchment exported 1 545 kg N ha$^{-1}$ from agricultural areas. The cumulated N-input from the entire catchment (measured downstream) from 1976 to 2015 (overlapping in- and output) was 42 758 t, while the riverine export in the same time was only 15 % implying an

10  agricultural export of 397 kg N ha$^{-1}$ (Fig. 5). This mass discrepancy between in- and output translates into a retention rate in the entire Holtemme catchment of 85 %. The missing N is either removed via denitrification or is still being stored within the terrestrial system in the soil as biogeochemical legacy, or in soil water and groundwater as hydrological legacy. In relation to the entire subcatchment area (not only agricultural land use), the annual retention rate of $NO_3$-N was 10 kg N ha$^{-1}$ a$^{-1}$ in the midstream subcatchment and 55 kg N ha$^{-1}$ a$^{-1}$ in the flatter and more intensively cultivated downstream subcatchment.





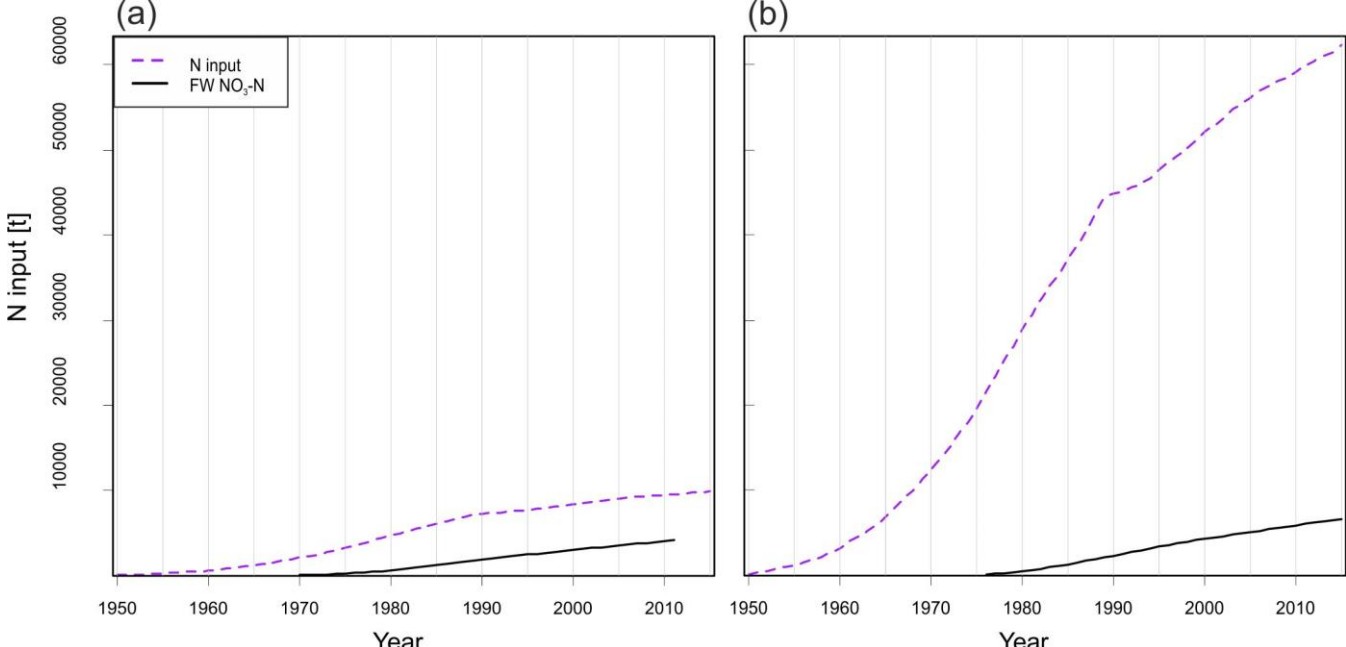

**Figure 5: Cumulative annual diffuse N-inputs to the catchment and measured cumulative NO$_3$-N exported load over time for Midstream (a) and Downstream (b).**

5 **3.4 Effective travel times of N**

With the fitted log-normal distribution for all seasonal concentration trajectories at the midstream and downstream stations, we were able to approximate the effective TTDs of NO$_3$-N through the catchments (Fig. 6; Table 4). Note that the upstream station was not used for this approach as no sufficiently resolved input signal (linear input increase between 1950 and 1979) was available. In general, the optimized distributions were able to sufficiently capture the time lag and smoothing between
10 the input and output concentrations (R$^2$ ≥ 0.83; S2.1; S2.2). Systematic differences between stations and seasons can be observed, best represented by the mode (peak TT).



**Table 4: Best fit parameters of the log-normal effective travel time distribution of N.**

|  | Parameter | All seasons | Winter | Spring | Summer | Fall |
|---|---|---|---|---|---|---|
| Midstream | μ | 2.8 | 2.8 | 2.6 | 2.8 | 3.0 |
|  | σ | 0.5 | 0.6 | 0.7 | 0.5 | 0.4 |
|  | Mode [a] | 12.5 | 11.6 | 7.8 | 13.5 | 17.0 |
|  | $R^2$ | 0.91 | 0.86 | 0.87 | 0.93 | 0.86 |
| Downstream | μ | 2.8 | 3.0 | 2.6 | 2.7 | 2.9 |
|  | σ | 0.6 | 0.6 | 0.8 | 0.4 | 0.5 |
|  | Mode [a] | 11.9 | 14.2 | 7.5 | 12.8 | 14.3 |
|  | $R^2$ | 0.96 | 0.9 | 0.83 | 0.93 | 0.85 |

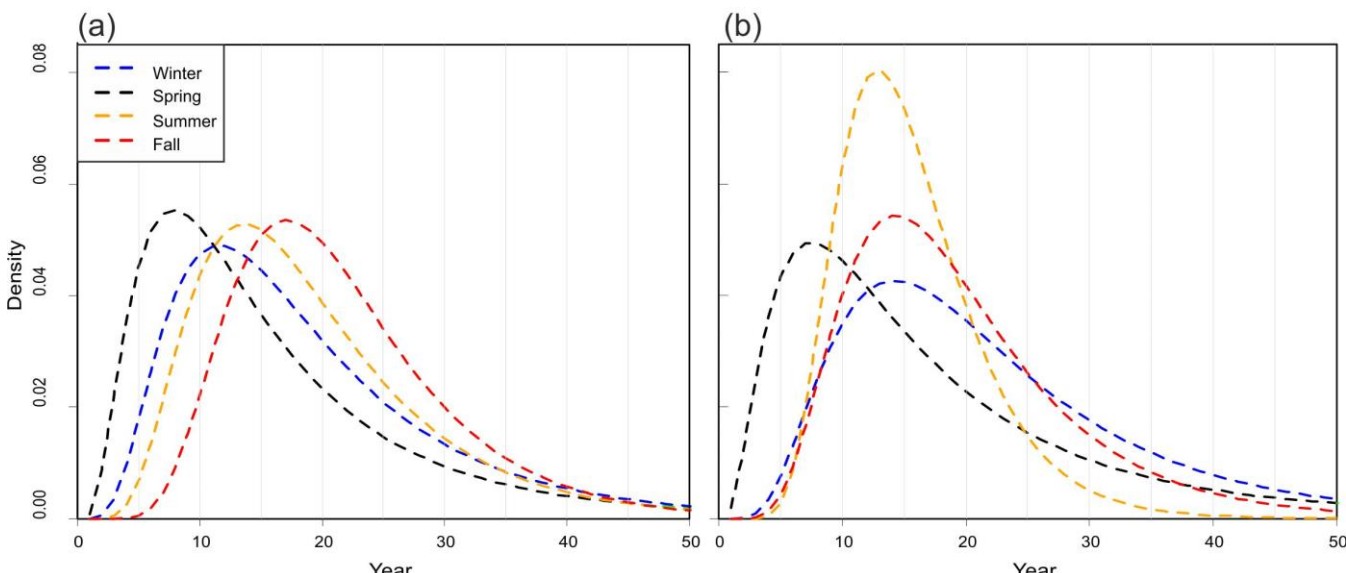

**Figure 6: Fitted log-normal distributions of effective travel times of N for Midstream (a) and Downstream (b).**

The TTs for all seasons taken together were almost identical for the mid- and downstream stations. However, the comparison of modes for the different seasons Midstream showed distinctly differing peak TTs between 8 a (spring) and 17 a (fall), which represented more than a doubling of the peak TT. Fastest times appeared in the HFSs while modes of the TTDs appeared longer in the LFSs. Note that the shape factor σ of the effective TTs also changed systematically: The HFs spring
10 and winter exhibited generally higher shape factors than the LFSs. This refers to a change in the Midstream coefficient of variation of the distributions from 0.8 in spring to 0.4 in fall.





The modes of the fitted functions for the Downstream station during the HFSs (8 a in spring, 14 a in winter) were almost identical to the ones at the midstream station. Conversely, fall exhibited shorter TTs for the downstream station than for the midstream station. The mode of the TTs ranged between 8 a (spring) and 14 a (winter, fall). Also the shape factors downstream ranged between 0.8 (spring) and 0.4 (summer). In summary, HFSs in both subcatchments had quite similar

5    TTDs, whereas the LFSs showed distinct differences in their peak time.

### 3.5 Seasonal nitrate concentrations and C–Q relationships over time

As described above, the Holtemme catchment showed a pronounced seasonality in discharge conditions, producing a HFS in December–May (winter + spring) and LFS in June–November (summer + fall). Therefore, changes in the seasonal concentrations of $NO_3$-N can also be associated with changes in the annual C–Q relationship.

10   In addition to the described changes in the N-output on annual time scales, also changing seasonal dynamics were quite common in the data record and can provide more detailed information about N-trajectories in the Holtemme catchment.





**Figure 7: Annual N-input (referred to the whole catchment, 2nd y-axis) to the catchment and measured median NO₃-N concentrations in the stream (1st y-axis) over time at three different locations. Upstream (a1), Midstream (b1), Downstream (c1). Plot of slope b vs. $CV_C/CV_Q$ for NO₃-N for the three catchments. X-axis gives the coefficient of variation of C relative to the coefficient of variation of Q. Y-axis gives the slope b of the linear ln C–ln Q-relationship. Colours indicate the temporal progression from 1970–2016 starting from red to yellow. Upstream (a2), Midstream (b2), Downstream (c2).**

In the pristine upstream catchment, no changes in the seasonal differences of riverine NO₃-N concentrations could be found (Fig. 7a1). The C–Q relationship showed a steady accretion pattern with highest concentrations in the HFSs winter and spring. The ratio of $CV_C/CV_Q$ changed only marginally (amplitude of 0.2) over time.

At the midstream station (Fig. 7b2), the early 70s showed the same seasonality as in the upstream catchment, but with a general increase of concentrations from 1970–1995. During the 80s, the increase of concentrations in the HFS was steeper/faster than in the LFS, which changed the seasonality to a strongly positive pattern ($b_{max}$=0.42, 1987) between C and Q. This development in the 80s was characterized by intra-annual amplitudes ($C_{spring} − C_{fall}$) of up to 2.4 mg L⁻¹ (1987),





which was a tripling within the years. With a lag of around 10 years, in the 90s, also the LFSs showed increased concentrations ($c_{max}$= 3.1 mg L$^{-1}$, 1998). These two peaks in the 80s in the HFSs and in the 90s in the LFSs cause bimodality in the concentration time series. The C–Q relationships showed a trisection evolving from an intensifying accretion pattern in the 70s and 80s to a constant pattern between C and Q in the 90s and afterwards. The $CV_C/CV_Q$ increased during the first

decade and decreased afterwards strongly by 0.4 between 1984 and 1995, showing a trajectory from a more chemodynamic to a chemostatic regime.

At the downstream station (Fig. 7c1), the concentrations over time in the HFS proceeded like observed at the midstream station, but with a much more pronounced decrease from 2010 on. The intra-annual amplitude in the 80s showed a maximum of 2.4 mg L$^{-1}$ (1983/84), with strongly positive C–Q patterns ($b_{max}$= 0.4; 1985). Again with a time lag, also concentrations

during LFSs peaked. Deviating from the bimodal concentration trends in the mid- and downstream HFSs, the LFSs downstream showed an unimodal pattern peaking in 1995/96 with concentrations above 6 mg L$^{-1}$ NO$_3$-N ($c_{max}$=6.9 mg L$^{-1}$). This increase in concentrations in LFSs above the concentrations in the HFSs caused dilution C-Q patterns in the 90s. This unimodal concentration trajectory in its shape and amplitude in the LFSs is unique at the downstream station and cannot be found in the other stations. Therefore, it can be stated that the seasonality did change with time, but as well as over space.

Due to the decline of low flow concentrations after 1995, the dilution pattern evolved to a constant C–Q pattern from the 00s onward. The $CV_C/CV_Q$ declined by 0.7 between 1984 (max. $CV_C/CV_Q$) and 2003 (min. $CV_C/CV_Q$) evolving to chemostatic export.

Despite the differences in concentrations, the trajectory of concentrations between Midstream and Downstream (bimodal vs. unimodal), the overall pattern in the C–Q relationship proceeding from accretion pattern to constant C–Q relationships were

the same in both agriculturally used subcatchments (Fig. 7a2–c2), while the pristine catchment showed no changes of the intra-annual seasonality. However, at the downstream station, an additional dilution pattern was observed for several years. In both managed subcatchments, the temporal concentration trajectory was accompanied by a dominantly decreasing $CV_C/CV_Q$ ratio evolving from chemodynamic to chemostatic behavior.

## 4 Discussion

### 4.1 Catchment scale N-budgeting

Based on the calculated differences between N-inputs and N-outputs (via discharge) for the three subcatchments within the Holtemme catchment, we will discuss two potential reasons for the residual in the N-budget: 1) permanent N-removal by denitrification or 2) the build-up of legacies.

As described above, the load stemming from the most upstream, pristine catchment accounted for <10 % of the exported

annual load over the entire time period. This minor contribution can be attributed to the lack of agricultural and urban land use as dominant sources for N. Consequently N-export from the upstream subcatchment was assumed to be dominantly controlled by atmospheric deposition of N. As the cumulated export over the observation period was higher than the





assumed input, the estimation of a retention potential was difficult. This can be explained by unaccounted N-sources, e.g. stemming from minor anthropogenic activity. Moreover, the assumed constant biological N-fixation as described by Cleveland et al. (1999), may lead to an underestimation of the real N-input. The total input to the whole catchment area was quantified with almost 43 000 t (1976–2015) and compared to the respective output over the same time period yielded export

rates of 46 % at the midstream and 15 % at the downstream station (Table 3), respectively. The reasons for the difference in export rates between the two subcatchments can be various. The most likely ones, differences in discharge, topography and denitrification capacity, will be discussed in the following.

Load export of N from agricultural catchments is assumed to be mainly discharge-controlled (Basu et al., 2010). Many solutes show a lower variance in concentrations compared to the variance in stream flow, which makes the flow variability a

strong surrogate for load variability (Jawitz & Mitchell, 2011). This can also be seen in the Holtemme catchment, which evolved to a more chemostatic export regime over time (Fig. 7). Highest N-export and lowest retention were observed in the midstream subcatchment, where the highest discharge contribution can be found.

Besides discharge-quantity, we argue that the expected midstream flow paths favor a fast leaching of $NO_3$-N. The higher percentage of artificial drainage by tiles and ditches (59 % vs. 21 %; S1.1) as well as the steeper slopes (3.2 vs. 1.9) in the

non-forested area of the midstream catchment, promote rapid, shallow subsurface flows. These flow paths can more directly connect agricultural N-sources with the stream and in turn cause elevated instream $NO_3$-N concentrations (Yang et al., 2018). Related to surface topography, steeper terrain suggests a deeper vertical infiltration (Jasechko et al., 2016) and also a leaching of $NO_3$ from a wider depth-range than flat terrains. Vice versa, fewer drainage installations, less slope and shallower discharge contribution could decrease the export (increase the retention) potential downstream.

The only process able to permanently remove N-input from the catchment is denitrification in soils, aquifers (Seitzinger et al., 2006; Hofstra & Bouwman, 2005) and at the stream-aquifer interface such as in the riparian (Vidon & Hill, 2004; Trauth et al., 2018) and hyporheic zones (Vieweg et al., 2016). As the riverine exports are signals of the catchment or subcatchment processes, integrated in time and space, separating legacy of $NO_3$ from removal via denitrification is difficult. A clear separation of these two key processes, however, would be important for decision makers as both have different implications

for management strategies and different future impacts on water quality. Even if extensive groundwater quality measurements were available, using this more local type of information to for an effective catchment scale estimation of N-removal by denitrification would be challenging (Green et al., 2016; Otero et al., 2009; Refsgaard et al., 2014). Therefore we discuss the denitrification potential in the soils and in the groundwater of the Holtemme catchment based on a literature review. On the basis of isotopic compositions in the Holtemme River, a previous study (Müller et al., 2018) stated that

denitrification played no or only a minor role in the catchment. However, we still see the need to carefully check the potential of denitrification to explain the input-output imbalance considering other studies.

If 85 % of the N-input (42 758 t, dominantly agricultural input) to the catchment between 1976 and 2015 (39 a) were denitrified in the soils of the agricultural area (161 km²), it would need a rate of 57.9 kg N ha$^{-1}$ a$^{-1}$. Considering the derived TTs, denitrification of the convolved input would need the same rate (58 kg N ha$^{-1}$ a$^{-1}$, 1976–2015). Denitrification rates in



soils for Germany (NLfB, 2005) have been reported to range between 13.5–250 kg N ha$^{-1}$ a$^{-1}$, while rates larger than 50 kg N ha$^{-1}$ a$^{-1}$ may be found in carbon rich and waterlogged soils in the riparian zones near rivers and in areas with fens and bogs (Kunkel et al., 2008). As water bodies and wetlands make up only 1 % of the catchment´s agricultural land use in our catchment (Fig. 1; EEA, 2012), and consequently the extent of waterlogged soils is negligible, denitrification rates <50 kg N

ha$^{-1}$ a$^{-1}$ can be assumed. This contradicts the necessary rate needed to explain the retention by denitrification in soils only. Seitzinger et al. (2006) assume a rate of 14 kg N ha$^{-1}$ a$^{-1}$ for agricultural soils at a global scale. This could denitrify 24 % of the retained (85 %) catchment's N-input. Another study estimates for the soils of the Holtemme catchment very low to low denitrification rates, of 9–13 kg N ha$^{-1}$ a$^{-1}$ on the basis of a simulation with the modeling framework GROWA-WEKU-MEPhos (Kuhr et al., 2004). Hence, denitrification in the soils, including the riparian zone, may partly explain the retention

of NO$_3$-N, but is unlikely a single explanation for the observed imbalance between in- and output.

Regarding denitrification in groundwater, the literature provides denitrification rate constants of a first order decay process between 0.01–0.56 a$^{-1}$ (van Meter et al., 2017b; van der Velde et al., 2010; Wendland et al., 2005). We derived the denitrification constant by distributing the input according to the fitted log-normal distribution of TTs assuming a first order decay along the flow paths (Kuhr et al., 2004; Rode et al., 2009; van der Velde, 2010). The denitrification of the 85 % of

input mass would require a rate constant of 0.12 a$^{-1}$. This constant is in the range of values reported by mentioned modelling studies. However, Kuhr et al. (2004) exclude any denitrification in the upper aquifer for the Holtemme catchment in their modeling framework GROWA-WEKU-MEPhos. The large range of reported denitrification constants in the literature clearly calls for a more rigorous differentiation of denitrification in streams, groundwater and soils in future work. In this present study, however, the role of denitrification in groundwater to explain the observed imbalance between in- and output

cannot ultimately be quantified. .

Lastly, assimilatory NO$_3$ uptake in the stream may be a potential contributor to the difference between in- and output. But even with maximal NO$_3$ uptake rates as reported by Mulholland et al. (2004; 0.14 g N m$^{-2}$ d$^{-1}$) or Rode et al. (2016; max. 0.27 g N m$^{-2}$ d$^{-1;}$ estimated for a catchment adjacent to the Holtemme), the annual assimilatory uptake in the river would be a minor removal process, estimated to contribute only 3.2 % of the 85 % discrepancy between in- and output. Also

denitrification in the stream can be excluded as a dominant removal process. According to the rates reported by Mulholland et al. (2008; max. 0.24 g N m$^{-2}$ d$^{-1}$), the Holtemme River would need a 35-times larger area to be able to denitrify the retained N.

In summary, the precise differentiation between legacy and denitrification is not fully resolvable on the basis of the available data. Also a mix of both could account for the missing 85 % (Downstream) or 54 % (Midstream) in the N-output. Input-

output assessments with time series from different catchments, as presented in van Meter & Basu (2017), covering a larger variety of catchment characteristics, hold promise for an improved understanding of the controlling parameters and dominant retention processes.

The fact that current NO$_3$ concentration levels in the Holtemme River still show no clear sign of a significant decrease, calls for a continuation of the NO$_3$ concentration monitoring, best extended by additional monitoring in soils and groundwater.



Despite strong reductions in agricultural N-input since the 90s, the annual N-surplus (e.g. 818 t a$^{-1}$, 2015) is still much higher than the highest measured export (load$_{max}$ = 216 t a$^{-1}$, 1995) from the catchment. Hence, the difference between in- and output is still high and covering a factor of 4 during the past 10 a (factor of 5 with the shifted input according to 12 a of TT). Consequently, either the legacy of N in the catchment keeps growing instead of getting depleted or the system relies on a

potentially limited denitrification capacity. Denitrification may irreversibly consume electron donors like pyrite for autolithotrophic denitrification or organic carbon for heterotrophic denitrification (Rivett et al., 2008; Kunkel et al., 2008). Neither tolerating the build-up of legacies nor relying on denitrification represents sustainable and adapted agricultural management practice. Hence, also future years will face increased NO$_3$-N concentrations and loads exported from the Holtemme catchment.

**4.2 Linking effective TTs, concentrations and C–Q trajectories**

Based on our data-driven analyses we propose the following conceptual model (Fig. 8) for N-export from the Holtemme catchment, which is able to plausibly connect and synthesize the available data and findings on TTs, concentration trajectories and C-Q relationships.

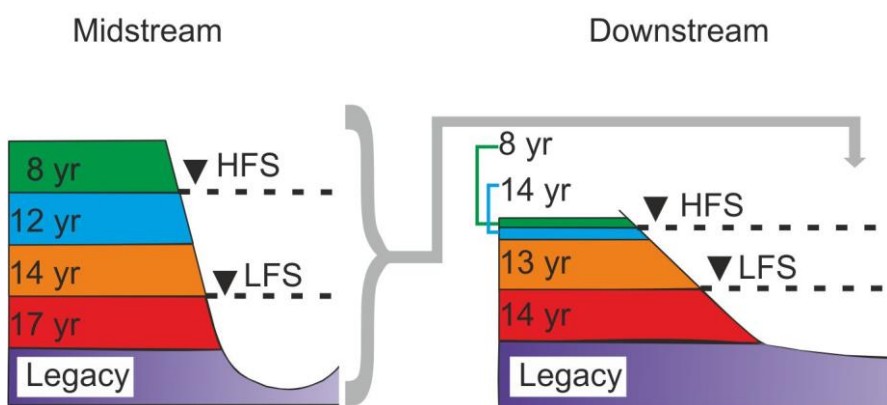

**Figure 8: Conceptual model. Hypothetical intra-annual Q contribution (with peak TTs) in different depths and changing water levels (black triangles) during LFS and HFS. The colour of the boxes refers to the seasons as used in Fig. 7.**

Over the course of a year, different subsurface flow paths are active, which connect different subsurface N-source zones with

different source strength (in terms of concentration and flux) to the streams. These flow paths transfer water and NO$_3$-N to the streams, predominantly from shallower parts of the aquifer when water tables are high during HFSs and exclusively from deeper groundwater during low flows in LFSs (Rozemeijer & Broers, 2007; Dupas et al., 2016; Musolff et al., 2016). This conceptual model allows us to explain the observed intra-annual concentration patterns and the distinct clustering of TTs into low flow and high flow conditions. Furthermore, it can explain the mobilization of nutrients from spatially distributed NO$_3$-

N sources by temporally varying flow-generating zones (Basu et al., 2010). Spatial heterogeneity of solute source zones can





be a result of downward migration of the dominant $NO_3$-N storage zone in the vertical soil-groundwater profile (Dupas et al., 2016). Moreover, a systematic increase of the water age with depths would, if denitrification in groundwater takes place uniformly, lead to a vertical concentration decrease. However, assuming that flow contributions from the same depths do not change between the years, the observed decadal changes in the seasonal concentrations cannot be explained by a stronger

imprint of denitrification with increasing water age. Under such conditions one would expect a steady seasonality in concentrations over time with $NO_3$-N concentrations that are always similarly high in HFSs and similarly low in LFSs, which we do not see in the data. Additionally, previous findings have indicated no or only a minor role of denitrification in the catchment (Kunkel et al., 2008; Müller et al. 2018). We instead argue that the vertical migration of a temporally changing $NO_3$-N input is the key explanation for our observations with regards to N-budgets, concentrations and C–Q trajectories.

At the midstream station the fast TTs during HFSs are assumed to be dominated by discharge from the shallowest source zone. This zone is responsible for the fast response of instream $NO_3$-N concentrations to the increasing N-inputs (70s to mid-80s). This fast lateral transfer especially in spring (shortest TT) may be also triggered by the presence of artificial drainage structures such as tiles and ditches. In line with the longer TTs, low flow $NO_3$-N concentrations were less impacted in the 70s to mid-80s as deeper source zones were still less affected by anthropogenic inputs. With ongoing time and a downward

migration of the high $NO_3$-N inputs (before 1990), also those deeper layers delivered increased concentrations to the stream (90s). In line with the increasing low flow concentrations (in the 90s) were the decreasing spring concentrations of $NO_3$ caused by a depletion of the shallower $NO_3$-N stocks and a downward migrating peak zone (see also Dupas et al., 2016). This depletion of the stock was a consequence of drastically reduced N-input after the German peaceful reunification in 1989. The bimodality in concentrations over time in all four seasons underlined the changing intra-annual dominance of vertically

activated zones.

This conceptual model of N-trajectories is additionally supported by the changing C–Q relationship over time. The seasonal cycle started with increasing $NO_3$-N-maxima during high flows and minima during low flows, since firstly shallow source zones were getting loaded with $NO_3$. Consequently, the accretion pattern was intensified in the first decades. The resulting positive C–Q relationship on a seasonal basis was found in many agricultural catchments worldwide (e.g. Aubert et al., 2013;

Martin et al., 2004; Mellander et al., 2014; Rodriguez-Blanco et al., 2015; Musolff et al. 2015). However, after several years of deeper migration of the N-input, the catchment started to exhibit a chemostatic $NO_3$-N export regime (after 90s), which was manifested in the decreasing $CV_c/CV_Q$ ratio. This stationarity could have been caused by a vertical equilibration of $NO_3$-N concentrations in all seasonally activated depth zones of the soils and aquifers after a more stable long-term N-input after 1995. According to the 50[th] percentile of the derived TT, after 16 a only 50 % of the input had been released Midstream.

Therefore without any strong changes in input, the chemostatic conditions caused by the uniform, vertical $NO_3$-N contamination will remain. At the same time, this chemostatic export regime supports the hypothesis of a legacy rather than denitrification as dominant reason for the imbalance between in- and output.

At the downstream station, the riverine $NO_3$ concentrations during high flows were dominated by midstream discharge, which explains the similarity with the midstream bimodality in concentrations as well as the comparable TTs. During low



flows, the contribution of the downstream subcatchment can contribute much more to the overall nitrate export. For the LFSs, we observed a higher $NO_3$-N concentration with a unimodal trajectory and shorter TTs compared to the midstream subcatchment. We argue that the lowland subcatchment supports higher water levels and thus faster TTs during low flows. Greater prevalence of young streamflow in flatter terrain was also described by Jasechko et al. (2016). But besides the earlier

peak time during low flows, the concentration was found to be much higher than midstream. To cause such high intra-annual concentration changes, the downstream $NO_3$-N load contribution, e.g. during the concentration peak 1995/96, had to be high: the summer season was 27 t, which is more than twice the median contribution (11 t). The effective export from the downstream catchment happened mainly during LFSs, which is also supported by the narrower TTD in the summer (Fig. 6). The difference between the $75^{th}$ and $25^{th}$ percentiles (7 a) was also the fastest of all seasons in the summer at the downstream

station. This could be one reason for the high concentrations in comparison to the midstream catchment and HFSs.

In contrast to the midstream catchment, the C–Q trajectory in the downstream catchment evolved from an enrichment pattern, dominated by the high concentration during high flows from Midstream to a dilution pattern, when the high concentrations in the LFS from the downstream subcatchment dominated. Although the low flow concentrations were slowly decreasing in the 00s and 10s, also the downstream catchment evolved to a chemostatic $NO_3$ export regime as Midstream

(Fig. 7).

Our findings support the evolution from chemodynamic to chemostatic behavior in managed catchments, but also emphasize that changing inputs of N into the catchment can lead to fast changing export regimes even in relatively slowly reacting systems. Our findings expand on previous knowledge as we could show systematic inter-annual C–Q changes that are in line with a changing input and a systematic seasonal differentiation of TTs. Although our study showed chemostatic behavior

towards the end of the observation period (Mid- and Downstream; Fig. 7), this export regime is not necessarily stable as it depends on a continuous replenishment of the legacy store. Changes in the N-input translated to an increase of spatial heterogeneity in $NO_3$-N concentrations in soil- and groundwater with contrasting water ages. The seasonal changing contribution of different water ages thus results in more chemodynamic $NO_3$-N export regimes. As described in Musolff et al. (2017) both, export regimes and patters are therefore controlled by the interrelation of travel time and source

concentrations. We argue that a hydrological legacy of $NO_3$-N in the catchment has been established that resulted in a pseudo-chemostatic export behavior we observe nowadays. We furthermore argue that a biogeochemical legacy corresponding to the build-up of organic N in the root zones of the soil (van Meter et al., 2016) is less probable. If we assume that all of the 84 % of the N-input is accumulating in the soils, we cannot explain the observed shorter-term inter-annual concentration changes and trajectory in the C–Q relationships. We would rather expect a stronger and even growing

dampening of the N-input to the subsurface with the built-up of a biogeochemical legacy in form of organic nitrogen. However, we cannot fully exclude the accumulation of a protected pool of soil organic matter with very slow mineralization rates as described in van Meter et al. (2017). Our conceptual model assigns the missing N to the long TTs of $NO_3$-N in soil- and groundwater and in turn to a pronounced hydrological legacy. In the midstream subcatchment, the estimated TTD explains 58 % of the retained $NO_3$-N, comparing the convolution of TTD with the N-input time series to the actual riverine



export. The remaining 42% cannot be fully explained at the moment and may be assigned to a permanent removal by denitrification (see discussion above), to a fixation due to biogeochemical legacy, or to more complex e.g. longer tailed TTDs, which are not well represented by our assumed log-normal distribution. In the downstream subcatchment, our approach explains 31 % of the observed export. This could in principle be caused by the same processes as described for the midstream subcatchment. However, in the downstream subcatchment we assume a hydrological legacy store in deeper zones without significant discharge contribution (Fig. 8). That mass of N is either bypassing the downstream monitoring station (note that the downstream station is still 3 km upstream of the Holtemme catchment outlet) or is affected by a strong time delay and dampening not captured by our approach. Consequently, future changes in N-inputs will also change the future export patterns and regimes, since this would shift the homogeneous $NO_3$-N distributions in vertical soil and groundwater profiles back to more heterogeneous ones.

## 5 Conclusion

In the present study, we used a unique time series over four decades from a mesoscale catchment as well as estimated N-input and discussed the linkage between the two on annual and intra-annual time scales. From the input-output assessment, the build-up of a potential N-legacy was quantified, effective TTs of nitrate were estimated and the temporal evolution to chemostatic $NO_3$-N export was investigated. This study provides four major findings.

First, the retention capacity of the catchment for N is 85 % of the N-input (input and output referring to 1976 to 2015), which can either be stored as a legacy or denitrified in the terrestrial or aquatic system. Although we could not fully quantify denitrification, we argue that this process is not the dominant one in the catchment to explain input-output differences. The observed N-retention can be more plausibly explained by legacy than by denitrification. In consequence, the hydrological N-legacy, i.e. the load of nitrate still on the way to the stream, may have strong effects on future water quality and long-term implications for river water quality management. With a median export rate of 162 t $a^{-1}$ (1976–2016, downstream), a depletion of this legacy (<36 000 t N) via baseflow would maintain elevated riverine concentrations for the next decades. Although N-surplus strongly decreased after the 80s, during the past 10 a there still was, an imbalance between input and export by a mean factor of 5 (assuming the temporal offset of peak TTs between in- and output of 12 a). This is a non-sustainable condition, regardless of whether the retained nitrate is stored or denitrified.

Secondly, we derived peak time lags between N-input and riverine export between 9–17 a that systematically differ seasonally. Catchment managers should be aware of these long time frames when implementing measures and when evaluating them. This study explains the seasonally differing lag times and temporal concentration evolutions with the vertical migration of the nitrate and their changing contribution to discharge by seasonally changing aquifer connection. Hence, inter-annual concentration changes are not dominantly controlled by inter-annually changing discharge conditions, but rather by the seasonal changing activation of subsurface flows with differing ages and thus differing N-loads. As a consequence of this activation-dependent load contribution, an effective, adapted monitoring needs to cover, different



discharge conditions when measures shall be assessed for their effectiveness. Thus, there is a general need for sufficient monitoring length and appropriate methods for data evaluation like the seasonal statistics of time series.

Third, in contrast to a more monotonic change from a chemodynamic to a chemostatic nitrate export regime that was observed previously (Dupas et al., 2016; Basu et al., 2010), this study found a systematic change of the nitrate export regime

from accretion over dilution to chemostatic behavior. Here we can make use of the unique situation in East-German catchments where the collapse of agriculture in the early 90s provided a large scale "experiment" with abruptly reduced N-inputs. While previous studies could not distinguish between biogeochemical and hydrological legacy to cause chemostatic export behavior, our findings support a hydrological legacy. The systematic inter-annual changes of C-Q relationships of $NO_3$-N was explained by the changes in the N input in combination with the seasonally changing effective travel times of N.

The observed export regime and pattern of $NO_3$-N helped to define the dominance of a hydrological N-legacy over the biogeochemical N-legacy in the upper soils. In turn, observed trajectories in export regimes of other catchments may be an indicator of their state of homogenization and can be helpful to classify results and predict future concentrations. Only on the basis of long-term time series these inter-annual systematic changes in C–Q relationships can be detected.

Fourth, although we observed long TTs (slow catchment reaction), significant input changes also showed strong inter-annual

changes in the export regime. Chemostatic behavior is therefore not necessarily a persistent endpoint of intense agricultural land use, but depends on steady replenishment of the N-store. Therefore, the export behavior can also be termed pseudo-chemostatic and may further evolve in the future (Musolff et al., 2015) under the assumptions of a changing N-input. Depending on the size of the legacy, a significant reduction or increase of N-input can cause an evolution back to dilution or enrichment patterns. Simultaneously, input changes affect the homogenized vertical nitrate profile, resulting in larger intra-

annual concentration differences and consequently chemodynamic behavior. Hence, chemostatic behavior and homogenization are characteristics of managed catchments, but only under constant N-input.

Recommendations for a sustainable management of nitrogen in the Holtemme catchment, also transferable to comparable catchments, focus on the three aspects.

- Our findings could not prove a significant loss of $NO_3$-N by denitrification. To deal with the past inputs and to

focus on the depletion of the N-legacy, end-of-pipe measures such as riparian buffers or constructed wetlands may initiate N-removal by denitrification (Messer et al., 2012).

- We could show that there is still an imbalance of N-input and riverine export by a factor of 6. A reduced N-input due to better management of fertilizer and the prevention of N-losses from the root zone in present time is indispensable to enable depletion instead of a further build-up or stabilization of the legacy.

- For future times, we should utilize a much wider range of catchments with long-term observations for understanding the spatial and temporal variation of legacy build-up, denitrification and TTs as well as their controlling factors. Data-driven analyses of differing catchments covering a higher variety of characteristics may provide a more comprehensive picture of N-trajectories and their controlling parameters.





**Data availability**

Discharge data (for all dates) and water quality data (from 1993) can be accessed at the websites of the State Office of Flood Protection and Water Management (LHW) Saxony-Anhalt (http://gldweb.dhi-wasy.com/gld-portal/). Atmospheric deposition data between 1995 and 2015 can be accessed at the website of the Meteorological Synthesizing Centre - West

(MSC-W) of the European Monitoring and Evaluation Programme (EMEP) (http://www.emep.int/mscw/index_mscw.html) that is assigned to the Meteorological institute of Norway (MET Norway).

**Author contribution**

Sophie Ehrhardt, carried out the analysis, interpreted the data and wrote the manuscript.

Andreas Musolff designed the study and co-wrote the manuscript.

Rohini Kumar contributed discharge modelling results and atmospheric deposition and co-wrote the manuscript.

Jan Fleckenstein and Sabine Attinger contributed to the study design and helped finalizing the manuscript.

**Competing interests**

The authors declare no conflict of interest.

**Acknowledgement**

We would like to thank the State Office of Flood Protection and Water Management (LHW) Saxony-Anhalt for supplying data for discharge and water quality. Additionally, we thank the MET Norway, as source of data to model the atmospheric deposition. We gratefully acknowledge the provision of N-input from agriculture by Martin Bach, University of Gießen, Germany.

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
