# Peer review of "Trajectories of nitrate input and output in three nested catchments along a land use gradient"

_Hydrology and Earth System Sciences, 2018_

## Referee Comment (RC1) · Anonymous Referee #1 · 6 Nov 2018

This paper entitled, "Decadal trajectories of nitrate input and output in three nested catchments along a land use gradient" by Erhardt and others reports the result of a long-term nitrogen budget constrained with modeled and empirical estimates of N surplus and hydrological export. The authors found that most of the N was retained and attribute this mainly to hydrological legacy (storage within the catchment—primarily in the aquifer), based on a lack of observed denitrification in other studies, and the changes in the concentration-discharge relationships. The study is compelling and brings up many societally-urgent issues about preserving water quality in the Anthropocene. The paper is generally well written, though sections are uneven in their detail (either going too much into the weeds or not giving enough detail). I had a few major

questions and then provide line edits below. I believe that with a thorough revision, this article could be an important contribution to this journal. 1. How is error propagated? The authors often report four significant figures, but do not report standard deviation, confidence intervals, or some other estimate of uncertainty. Given the compound assumptions of the input chronicle models and the hydrological components, a sensitivity analysis or some kind of quantification of uncertainty seems warranted. 2. The idea of comparing biogeochemical and hydrological legacies is very compelling but it remains unclear to me how these parameters were estimated and compared. Structuring the methods around the research questions or overarching hypotheses and carrying this through the manuscript would make this flow clearer would make the results/discussion more impactful. 3. I think the discussion would be more engaging if the authors focused on the applicability of this approach to catchments generally, rather than explaining specific observations from their study. They do this effectively several times (e.g. starting on page 22 starting around line 20), but there is also quite a bit of retreatment of the results, which are specific to these sites. 4. The authors present an interesting puzzle of massive nitrogen retention/removal that cannot be attributed to typical pathways (e.g. denitrification, uptake, mineral association). The authors then conclude that N storage (the biogeochemical and hydrological legacies) account for the disconnect. However, the dismissal of denitrification seems to be based on a few studies from this area, which are not described in detail (e.g. Page 23, line 15). If these other studies are definitive and reliable, more description of their methods should be given. Another explanation is associated with point 1—could the N removal be much lower when uncertainty in inputs and outputs are included?

Line edits Page 2 Line 5: (Elser et al. 2007) Line 6: It seems odd to say these changes were strictly terrestrial. It seems they influenced both. Line 10: Do the authors mean the natural rate of reactive N fixation has been doubled (e.g. (Vitousek et al. 1997))? Page 3 Line 2: management interventions (instead of "measures")? Line 2: Recent study from similar agricultural and climatic context that found decadal hydrologic (Kolbe et al. 2016; Marçais et al. 2018) Line 16: I actually think there are quite a few studies,
especially recently (Dupas et al. n.d.; Howden et al. 2010; Burt et al. 2011; Minaudo et al. 2015; Meter & Basu 2017; Abbott et al. 2018; Coble et al. 2018; Garnier et al. 2018; Marcé et al. 2018; Pinay et al. 2018; Fanelli et al. 2019) Line 20: How do these analyses compare with soil-surface N balance approaches that include a crop and livestock removal component (Poisvert et al. 2017; Abbott et al. 2018)? Line 30: Recent paper on concentration-discharge responses to catchment saturation (Moatar et al. 2017) Page 5 Line 18: In what dimensions is this catchment especially vulnerable to climate change? Page 8 Line 13-20: Interesting that the primary datasets do not include non-agricultural land for N deposition. Why did the authors not use one of the products that provided a consistent N deposition rate across land-use types? Perhaps this is a small portion of the overall N budget, but it would be worthwhile to specify. Page 9 Figure 2: The dissimilarity in the NO3 concentration time series is striking as are the drops to zero mg/L even at the lowest site. Consider combining Figures 2 and 3 to allow visual comparison of discharge and concentration. Page 10 Line 9: the discharge time series were used. . . Page 11 Line 8: allows increasing . . . Page 12 Line 10: Because our purpose was to balance and compare . . . Line 12: This justification seems unclear. Is it simply claiming that the longer-term trends are accurate, though the daily values are not? Page 14 Table 2: These differences in specific discharge are remarkable. Is this typical for this area or is the three-fold difference due to a known environmental or anthropogenic variable? Page 15 Line 11: Revise sentence for grammar and clarity (with implications for instead of with discussion on?) Page 16 Line 14: It is striking that the retention capacity increases 5-fold with landscape position. Is this because of shifts in soil and subsurface properties or because the retention or removal rates are dependent on substrate concentration? Page 22 Line 20: Nitrification also results in gaseous N loss via the "leaky pipe" pathway (Hart et al. 1994). Line 29: Is this referring to denitrification in the near-surface zone or throughout the whole catchment? With pyrite, sulfur, and other iron ubiquitous in the weathered and fractured zones, aquifer denitrification is likely occurring Page 23 Line 18: New methods for constraining aquifer travel time to constrain removal rates using numerical or empirical methods (Kolbe et al.

[Figure]

2016; Marçais et al. 2018). Page 25 Line 1: Similar to these observations, though they are on a much smaller scale (Thomas & Abbott 2018) Page 28 Line 9: were explained Line 14: catchment reaction seems like an odd description for transit time.

Citations: Abbott, B.W., Moatar, F., Gauthier, O., Fovet, O., Antoine, V. & Ragueneau, O. (2018). Trends and seasonality of river nutrients in agricultural catchments: 18 years of weekly citizen science in France. Science of The Total Environment, 624, 845–858. Burt, T.P., Howden, N.J.K., Worrall, F. & McDonnell, J.J. (2011). On the value of long-term, low-frequency water quality sampling: avoiding throwing the baby out with the bathwater. Hydrological Processes, 25, 828–830. Coble, A.A., Wymore, A.S., Shattuck, M.D., Potter, J.D. & McDowell, W.H. (2018). Multiyear Trends in Solute Concentrations and Fluxes From a Suburban Watershed: Evaluating Effects of 100-Year Flood Events. Journal of Geophysical Research: Biogeosciences, 123, 3072–3087. Dupas, R., Minaudo, C., Gruau, G., Ruiz, L. & Gascuel‐Odoux, C. (n.d.). Multidecadal Trajectory of Riverine Nitrogen and Phosphorus Dynamics in Rural Catchments. Water Resources Research, 0. Elser, J.J., Bracken, M.E.S., Cleland, E.E., Gruner, D.S., Harpole, W.S., Hillebrand, H., et al. (2007). Global analysis of nitrogen and phosphorus limitation of primary producers in freshwater, marine and terrestrial ecosystems. Ecology Letters, 10, 1135–1142. Fanelli, R.M., Blomquist, J.D. & Hirsch, R.M. (2019). Point sources and agricultural practices control spatial-temporal patterns of orthophosphate in tributaries to Chesapeake Bay. Science of The Total Environment, 652, 422–433. Garnier, J., Ramarson, A., Billen, G., Théry, S., Thiéry, D., Thieu, V., et al. (2018). Nutrient inputs and hydrology together determine biogeochemical status of the Loire River (France): Current situation and possible future scenarios. Science of The Total Environment, 637–638, 609–624. Hart, S.C., Stark, J.M., Davidson, E.A. & Firestone, M.K. (1994). Nitrogen mineralization, immobilization, and nitrification. Methods of Soil Analysis: Part 2—Microbiological and Biochemical Properties, 985–1018. Howden, N.J.K., Burt, T.P., Worrall, F., Whelan, M.J. & Bieroza, M. (2010). Nitrate concentrations and fluxes in the River Thames over 140 years (1868–2008): are increases irreversible? Hydrological Processes, 24, 2657–2662. Kolbe, T., Marçais, J., Thomas,

[Figure]

Z., Abbott, B.W., de Dreuzy, J.-R., Rousseau-Gueutin, P., et al. (2016). Coupling 3D groundwater modeling with CFC-based age dating to classify local groundwater circulation in an unconfined crystalline aquifer. Journal of Hydrology, 543, Part A, 31–46. Marçais, J., Gauvain, A., Labasque, T., Abbott, B.W., Pinay, G., Aquilina, L., et al. (2018). Dating groundwater with dissolved silica and CFC concentrations in crystalline aquifers. Science of The Total Environment, 636, 260–272. Marcé, R., Schiller, D. von, Aguilera, R., Martí, E. & Bernal, S. (2018). Contribution of Hydrologic Opportunity and Biogeochemical Reactivity to the Variability of Nutrient Retention in River Networks. Global Biogeochemical Cycles, 32, 376–388. Meter, K.J.V. & Basu, N.B. (2017). Time lags in watershed-scale nutrient transport: an exploration of dominant controls. Environ. Res. Lett. Minaudo, C., Meybeck, M., Moatar, F., Gassama, N. & Curie, F. (2015). Eutrophication mitigation in rivers: 30 years of trends in spatial and seasonal patterns of biogeochemistry of the Loire River (1980–2012). Biogeosciences, 12, 2549–2563. Moatar, F., Abbott, B.W., Minaudo, C., Curie, F. & Pinay, G. (2017). Elemental properties, hydrology, and biology interact to shape concentration-discharge curves for carbon, nutrients, sediment, and major ions. Water Resour. Res., 53, 1270–1287. Pinay, G., Bernal, S., Abbott, B.W., Lupon, A., Marti, E., Sabater, F., et al. (2018). Riparian Corridors: A New Conceptual Framework for Assessing Nitrogen Buffering Across Biomes. Front. Environ. Sci., 6. Poisvert, C., Curie, F. & Moatar, F. (2017). Annual agricultural N surplus in France over a 70-year period. Nutrient Cycling in Agroecosystems, 107, 63–78. Thomas, Z. & Abbott, B.W. (2018). Hedgerows reduce nitrate flux at hillslope and catchment scales via root uptake and secondary effects. Journal of Contaminant Hydrology, 215, 51–61. Vitousek, P.M., Aber, J.D., Howarth, R.W., Likens, G.E., Matson, P.A., Schindler, D.W., et al. (1997). Technical Report: Human Alteration of the Global Nitrogen Cycle: Sources and Consequences. Ecological Applications, 7, 737.

---

## Referee Comment (RC2) · Anonymous Referee #2 · 23 Nov 2018

General comments: The manuscript addresses the important issue of legacy stores of nutrients, which may prevent mitigation actions that reduce the inputs from having immediate effects on stream water quality. I like the date drive approach to investigate the travel times of nitrate. The paper shows that 85% of the N input is retained within the catchment. The investigation about the fate of this lost N is not very convincing and inconclusive. Based on data on inputs and outputs alone, the authors cannot proof whether the N is retained in the soil, whether it is traveling along long flow paths, or whether it is denitrified. The authors try to give answers based on literature, but this is not very convincing. A weak point of the paper is that the entire soil and groundwater system is addressed as a black box. This is a bit strange given the focus on the

paper on N –stores and travel times in soil and groundwater. Including data on e.g. groundwater heads and flow-paths and concentration depth profiles for N could provide more certainty about the fate of the lost N.

Specific comments:

Title: Consider to leave out 'decadal'. I don't understand why you would only be looking at decadal trajectories

Abstract: The abstract is rather long. Especially the description of the results (from "We show..."). Consider to start a new paragraph here to make the structure more clear. The conclusion statement is a bit weak. Management should both address longer term and short term N-loads. How does this change water quality management in practice?

Introduction: From P3L4 until P4L22 the introduction reads like a description and a justification of the methods that you apply. It remains unclear what is not yet known from the existing scientific literature, why that is important, and what new science this paper brings. In P4L20 you state that "data-driven studies focused either solely on N-budgeting and legacy estimation or on TTs." What data-driven studies do you mean here? Why is this a problem / what problem do you solve by combining these? The referencing to Van Meter and Basu is quite excessive P2L11-12: here you state that the agricultural nitrogen input is still high since the 1980's. It did decrease in most EU member states since the '80s as a result of the introduction of manure legislation, didn't it? P2L26: "The evaluation of measures..." What evaluation of measures. This sentence is a bit hard to follow.

P5L18: why is the region vulnerable to climate change? P6L3: it's not clear where the 2 WWTP's are located. Can you add them to your map? P6L8: how much are agriculture and WWTP's (and other sources) contributing in %? Figure 1: the stream is not very clear on this map. P7L5 :"artificially drained" Do you mean drained by open ditches or by subsurface tube drains? How much has subsurface tube drainage? Table 1: The fraction artificially drained (last row) is much lower downstream. I would expect

more artificial drainage in the downstream part of the catchment as this is usually the wetter part of the catchment. Is there a reason why there is less artificial drainage needed in the downstream part? P8L30 "...we do not account for wastewater fluxes at this point..." Why is this legitimate? Is the wastewater N flux negligible? Figure 2 and 3: shouldn't these figures be presented in the results section? Figure 2c: It seems like the NO3 concentration is 0 around 2007 and at the end of the graph. Please check this. There also seems to be a regime-shift in this plot just before 2000. What happened? P11L6: "flow-normalized concentrations" It is not clear here why you need flow normalization. Consider to bring forward the end of the paragraph. Why would you want to take out the impact of variable flow conditions? P11L9: I don't understand how you interpolate the bi-weekly/monthly data. "...using a flexible statistical representation for every day of the discharge record". P13L14: "purple line"→purple dashed line P13L21: "peaked 1980" → peaked in 1980 Table 2: It is hard to connect the numbers for the LFS and HSF contributions in the text (<10%, 33%) with this table. It would be better not to give the cumulative contributions, so for HFS: 21, 69, 10. P15L11: I don't understand "...besides the statistical evaluation of the time series" P16L6-15: During the measurement period the catchment will partly export N-inputs from before 1970/76. This could be seen as the legacy of the period before the measurement period. The missing N described here adds to the legacy from before 1970/76. P18L6: why are these TTs for all seasons taken together not presented? Figure7b1: The concentrations seem to drop here, before the input drops. How is this possible? Figuret7c1: The higher concentrations in summer and fall during the peak around 1990 are surprising. This would indicate that the concentrations in deep groundwater with long travel times to surface water are higher than the concentrations in shallow groundwater with short travel times. Is this groundwater N that infiltrated in the midstream catchment and seeps up in the downstream catchment? Figure 7a2-c2: add a legend. P20L9: refer to figure 7a2. P20L7-P21L17: This text in combination with figure 7 is quite a hard puzzle. P22L1: "was difficult" → "was impossible" P22L1-2: Degradation of organic matter may play a role. P22L17-20: I don't understand why "steeper terrain suggests

a deeper infiltration" and "leaching of NO3 from a wider depth range than flat terrains". I would expect the opposite; deeper infiltration and leaching from a wider depth range in flat terrains. Of course, this depends on the geology. P22L26:"to for an" → "for an" P23L9-10: "Hence,...output" I think that this conclusion that denitrification is weakly supported by the previous text. Groundwater quality measurements would be very useful here. P23L16: why did Kuhr et al exclude denitrification? P24L4-9: from this paragraph and especially the last 2 sentences it seems like it is not important whether the legacy store is growing or the denitrification capacity is used, however on P22L23-25 you stated that this difference is important. Figure 8: This figure does not make any sense to me. P25L3-5: I don't think that you can make this assumption; the flow contributions from a certain depth can vary a lot due to interannual variability P26L3: You can also argue that groundwater seeping up is more important in the downstream catchment. This would mean more discharge of relatively old water.

---

## Author Comment (AC1) · 14 Jan 2019

General response: We thank the referee for the valuable inputs and remarks. We address all comments below and hope to clarify the questions raised.

1. How is error propagated? The authors often report four significant figures, but do not report standard deviation, confidence intervals, or some other estimate of uncertainty. Given the compound assumptions of the input chronicle models and the hydrological components, a sensitivity analysis or some kind of quantification of uncertainty seems warranted.

[Figure]

R1: Right, an uncertainty analysis was missing so far and will for sure improve the analysis. We suggest to give the mean standard error of the WRTDS regression analysis (for the modelled concentration) to provide this information. For the input of nitrogen, we refer to the methodological error provided by Bach & Frede (1998). We, moreover, will perform an uncertainty analysis for estimating effective travel times of N using a Monte-Carlo approach assuming normal distributed errors.

2. The idea of comparing biogeochemical and hydrological legacies is very compelling but it remains unclear to me how these parameters were estimated and compared. Structuring the methods around the research questions or overarching hypotheses and carrying this through the manuscript would make this flow clearer would make the results/discussion more impactful.

R2: That is a very helpful comment. We will revisit the research questions and make them more clear especially on where we see the potential of using the C-Q relationships to better disentangle the biogeochemical and hydrological legacies. Moreover, we will write an overview section for the method part to better integrate research questions with the method steps. Finally we will make the discussion on this topic more explicit in the discussion part as well. Based on the results of our analyses, we will improve upon the discussion part raising new hypothesis on dominant legacy types. Since our study is based on a data-driven analysis, we can't test such new hypothesis – but certainly we feel it is worth raising them from the data evidence so that a future imitative could start looking into those new aspects.

3. I think the discussion would be more engaging if the authors focused on the applicability of this approach to catchments generally, rather than explaining specific observations from their study. They do this effectively several times (e.g. starting on page 22 starting around line 20), but there is also quite a bit of retreatment of the results, which are specific to these sites.

R3: We will carefully review and revise the sections that are specific to our catchments

-potentially shorten them without losing the main message. Here we will more explicitly indicate where we discuss and draw conclusions for the studied catchment and where we can generalize our findings. We also see a greater potential of applying this local analysis to a wide range of catchments where we can more easily draw general conclusions (stated Page 28 Line 30-34).

4. The authors present an interesting puzzle of massive nitrogen retention/removal that cannot be attributed to typical pathways (e.g. denitrification, uptake, mineral association). The authors then conclude that N storage (the biogeochemical and hydrological legacies) account for the disconnect. However, the dismissal of denitrification seems to be based on a few studies from this area, which are not described in detail (e.g. Page 23, line 15). If these other studies are definitive and reliable, more description of their methods should be given. Another explanation is associated with point 1 - could the N removal be much lower when uncertainty in inputs and outputs are included?

R4: Yes – as given in R1 we will address the issue on uncertainty of the regression approach and the N input from agricultural areas. We would like to note here a recent paper published in November 2018 giving an overview on denitrification potential in the federal state this catchment is part of (Hannappel et al. 2018). It connects hydrochemical analysis of groundwater nitrate, oxygen and redox potential to the hydrogeological units in this region and states a general weak potential for denitrification for the study site. While revising our manuscript, we will include this study with methodological details to strengthen our argumentation on the denitrification part. It is however worth mentioning here that local groundwater information (which is also the base of this new paper by Hannappel et al. 2018) is hard to scale to the effective behavior at catchment scale. We have touched on this issue on Page 22, Lines 23-29. We already included the study by Müller et al. (2018) around the same study area that provided strong evidence on the lack of denitrification in their assessment based on isotopic signatures in the integrated nitrate signal in the surface water. We will put more emphasis on discussing this study as well to better argue our case in the revised manuscript.

[Figure]

5. Line edits Page 2 Line 5: (Elser et al. 2007)

R5: We will add this reference here.

6. Line 6: It seems odd to say these changes were strictly terrestrial. It seems they influenced both.

R6: We would drop the word "terrestrial" at the specified location.

7. Line 10: Do the authors mean the natural rate of reactive N fixation has been doubled (e.g. (Vitousek et al. 1997))?

R7: Yes, Vitousek et al. (1997) and Smil (1999) refer to the same: Human activities are mainly responsible for doubling the amount of reactive/ biological active N that enters the element's cycle from the unreactive atmospheric pool of N2. We will add these references and adjust the sentence, accordingly.

8. Page 3 Line 2: management interventions (instead of "measures")?

R8: Thanks – we will change that.

9. Line 2: Recent study from similar agricultural and climatic context that found decadal hydrologic (Kolbe et al. 2016; Marçais et al. 2018)

R9: We will consider these new studies.

10. Line 16: I actually think there are quite a few studies, especially recently (Dupas et al. n.d.; Howden et al. 2010; Burt et al. 2011; Minaudo et al. 2015; Meter & Basu 2017; Abbott et al. 2018; Coble et al. 2018; Garnier et al. 2018; Marcé et al. 2018; Pinay et al. 2018; Fanelli et al. 2019)

R10: Thanks for the input. We will check and consider these new studies.

11. Line 20: How do these analyses compare with soil-surface N balance approaches that include a crop and livestock removal component (Poisvert et al. 2017; Abbott et al. 2018)?

[Figure]

R11: Both, Jawitz and Mitchell (2011) as well as Musolff et al. (2015) are not based on N balances but on an interpretation of the temporal dynamic (or lack of temporal dynamic) in the observed nitrate concentrations. We will add that to this sentence. Our paper aims at a combination of both approaches – N balancing (since the N-input takes crops and livestock into account) and C-Q assessment.

12. Line 30: Recent paper on concentration-discharge responses to catchment saturation (Moatar et al. 2017)

R12: Thanks, we will refer to the suggested reference.

13. Page 5 Line 18: In what dimensions is this catchment especially vulnerable to climate change?

R13: A recent study by Wollschläger et al (2017) states a high vulnerability due to low water availability and a pronounced risk of summer droughts that is likely to be exacerbated by decreasing summer precipitation and increasing temperature/ potential evapotranspiration. Two new references can be included here as well (Marx et al. 2018, Samaniego et al. 2018). We will add this information in the revised manuscript.

14. Page 8 Line 13-20: Interesting that the primary datasets do not include non-agricultural land for N deposition. Why did the authors not use one of the products that provided a consistent N deposition rate across land-use types? Perhaps this is a small portion of the overall N budget, but it would be worthwhile to specify.

R14: We combine two products for N input to agricultural and non-agricultural land as there is no consistent product available in Germany, covering both with the required spatial and temporal precision. We will add this information to the text.

15. Page 9 Figure 2: The dissimilarity in the NO3 concentration time series is striking as are the drops to zero mg/L even at the lowest site. Consider combining Figures 2 and 3 to allow visual comparison of discharge and concentration.

R15: The "drops to zero" are actually the no-data-values that are erroneously displayed

as zero (but not considered in the WRTDS regressions). We will adjust the figure and also consider a combination with the discharge in Fig. 3.

16. Page 10 Line 9: the discharge time series were used. . . R16: Thanks – we will change this in the revised manuscript.

17. Page 11 Line 8: allows increasing . . . R17: Thanks – we will change this in the revised manuscript.

18. Page 12 Line 10: Because our purpose was to balance and compare . . . R18: Thanks – we will change this in the revised manuscript.

19. Line 12: This justification seems unclear. Is it simply claiming that the longer-term trends are accurate, though the daily values are not?

R19: No, the daily values are accurate but just that they not available at a daily time scale. We thus refer to the robust aggregated annual wastewater flux that much better fits to the flow normalized fluxes provided by the WRTDS regression analysis (see statement P12L10-14). Daily values are used to estimate an average fraction of NO3-N in the wastewater N flux.

20. Page 14 Table 2: These differences in specific discharge are remarkable. Is this typical for this area or is the three-fold difference due to a known environmental or anthropogenic variable?

R20: Yes this is remarkable but typical, and one of the reasons behind the establishment of the TERENO observatory system (Wollschläger et al. 2017). Wollschläger et al. (2017) state the strong precipitation gradient from 1700 mm/ a down to less than 500 mm/a within a range of 50 km due to the rain shadow of the Harz mountains. We will make a note on this in the revised manuscript.

21. Page 15 Line 11: Revise sentence for grammar and clarity (with implications for instead of with discussion on?)

R21: Thanks - we will revise the sentence.

22. Page 16 Line 14: It is striking that the retention capacity increases 5-fold with landscape position. Is this because of shifts in soil and subsurface properties or because the retention or removal rates are dependent on substrate concentration?

R22: Yes, this is quite a strong difference that is stated here as an observed result. Discussion on the reasoning can be found later on in Section 4.1.

23. Page 22 Line 20: Nitrification also results in gaseous N loss via the "leaky pipe" pathway (Hart et al. 1994).

R23: Right – there can be losses of N2O leaving the system at the nitrification step. However, in comparison to denitrification it does not appear to be a dominant loss term in N-budgets compared to denitrification (Rivett et al. 2008, Galloway et al. 2004). See also comment R4 – the paper by Müller et al. (2018) on the isotope evidence for the lack of N removal in our catchment.

24. Line 29: Is this referring to denitrification in the near-surface zone or throughout the whole catchment? With pyrite, sulfur, and other iron ubiquitous in the weathered and fractured zones, aquifer denitrification is likely occurring

R24: We refer to denitrification in general, taking both autotrophic and heterotrophic denitrification into account. Both need the absence of oxygen independent of whether electron donors are available or not. Also both affect the finally measured isotope signature in the remaining nitrate in the stream. See also our comment R4 with the new study (Hannappel et al. 2018) stating the lack of denitrification evidence that we will include in the manuscript.

25. Page 23 Line 18: New methods for constraining aquifer travel time to constrain removal rates using numerical or empirical methods (Kolbe et al. 2016; Marçais et al. 2018).

R25: Right. Enhanced knowledge on water travel time will improve the estimation of

reaction rates. We will consider the suggested studies.

26. Page 25 Line 1: Similar to these observations, though they are on a much smaller scale (Thomas & Abbott 2018)

R26: Thanks – we will consider this in the revised manuscript.

27. Page 28 Line 9: were explained R27: Thanks – we will make change as suggested.

28. Line 14: catchment reaction seems like an odd description for transit time.

R28: That is right. We will change that phrase as suggested.

References (that are not in the main manuscript): • Hannappel, S., Kopp, C. and Bach, T. (2018) Characterization of the denitrification potential of aquifers in Saxony-Anhalt. Grundwasser 23(4), 311-321. • Galloway, J.N., Dentener, F.J., Capone, D.G., Boyer, E.W., Howarth, R.W., Seitzinger, S.P., Asner, G.P., Cleveland, C.C., Green, P.A., Holland, E.A., Karl, D.M., Michaels, A.F., Porter, J.H., Townsend, A.R. and Vorosmarty, C.J. (2004) Nitrogen cycles: past, present, and future. Biogeochemistry 70(2), 153-226. • Marx, A., Kumar, R., Thober, S., Rakovec, O., Wanders, N., Zink, M., Wood, E.F., Pan, M., Sheffield, J. and Samaniego, L. (2018) Climate change alters low flows in Europe under global warming of 1.5, 2, and 3 degrees C. Hydrology and Earth System Sciences 22(2), 1017-1032. • Samaniego, L., Thober, S., Kumar, R., Wanders, N., Rakovec, O., Pan, M., Zink, M., Sheffield, J., Wood, E.F. and Marx, A. (2018) Anthropogenic warming exacerbates European soil moisture droughts. Nature Climate Change 8(5), 421. • Vitousek, P.M., Aber, J.D., Howarth, R.W., Likens, G.E., Matson, P.A., Schindler, D.W., Schlesinger, W.H. and Tilman, D. (1997) Human alteration of the global nitrogen cycle: Sources and consequences. Ecological Applications 7(3), 737-750.

---

## Author Comment (AC2) · 14 Jan 2019

General response: We thank the referee for the valuable inputs and remarks. We address all comments below and hope to clarify the questions raised.

General comments: 1. The manuscript addresses the important issue of legacy stores of nutrients, which may prevent mitigation actions that reduce the inputs from having immediate effects on stream water quality. I like the date drive approach to investigate the travel times of nitrate. The paper shows that 85% of the N input is retained within the catchment. The investigation about the fate of this lost N is not very convincing and inconclusive. Based on data on inputs and outputs alone, the authors cannot proof

whether the N is retained in the soil, whether it is traveling along long flow paths, or whether it is denitrified. The authors try to give answers based on literature, but this is not very convincing. A weak point of the paper is that the entire soil and groundwater system is addressed as a black box. This is a bit strange given the focus on the paper on N –stores and travel times in soil and groundwater. Including data on e.g. groundwater heads and flow-paths and concentration depth profiles for N could provide more certainty about the fate of the lost N.

R1: We understand the reviewer remark. Our study can only hypothesize and argue about travel times (TTs) and legacy, which is unfortunately a common methodological challenge of studies on catchment scale. We treat the entire catchment including soil and groundwater system as a black box and try to understand the inherent processes by looking at the signals produced or altered by this box. By doing such a data-driven analysis, our aim is to provide observation-based evidence on the system input-output response behavior, which can then be a starting point for developing either more targeted field-based or model-based "mechanistic" studies. Groundwater measurements and soil profiles would be a great help to support the hypothesis, but those observational records are generally not available. We tried hard to overcome this lack of knowledge with a comprehensive literature review for our studied catchment and comparable study sites (see also response R4 for Referee #1).

Specific comments: 2. Title: Consider to leave out 'decadal'. I don't understand why you would only be looking at decadal trajectories

R2: Agree, we will drop "decadal" in the title.

3. Abstract: The abstract is rather long. Especially the description of the results (from "We show: : :"). Consider to start a new paragraph here to make the structure more clear. The conclusion statement is a bit weak. Management should both address longer term and short term N-loads. How does this change water quality management in practice?

R3: We will carefully review the abstract, shorten the result section and put a clear focus on consequences for management practice that results from our analysis.

4. From P3L4 until P4L22 the introduction reads like a description and a justification of the methods that you apply. It remains unclear what is not yet known from the existing scientific literature, why that is important, and what new science this paper brings.

R4: We will carefully review the Introduction part of the paper to highlight the suggested mentioned aspects: What is known? Why it is important? And what's new in our study? Note that we have partly covered on these later on at P4, L18 – L33. But we will follow your advice and make it clearer the importance and the new scientific messages conveyed from our work.

5. In P4L20 you state that "data-driven studies focused either solely on N-budgeting and legacy estimation or on TTs." What data-driven studies do you mean here? Why is this a problem / what problem do you solve by combining these? The referencing to Van Meter and Basu is quite excessive.

R5: We refer to the data-driven studies e.g. by Worral et al., 2015 and Dupas et al., 2016 (as stated in P4, L19). We need to further underline the advantage of combining the quantification of legacy and TT in one study (and from the same data base) to use the TTs to explain the legacy. We will revise the text concerning this aspect. In terms of studies cited: See also comment R10 for Referee #1 where we aim at including a greater variety of studies.

6. P2L11-12: here you state that the agricultural nitrogen input is still high since the 1980's. It did decrease in most EU member states since the '80s as a result of the introduction of manure legislation, didn't it?

R6: The reviewer is right in pointing this out that N-inputs, also from agricultural sources were significantly reduced (but they are still on a high level). We will rewrite the concerned sentence to correct this inaccuracy.

[Figure]

7. P2L26: "The evaluation of measures: : :" What evaluation of measures. This sentence is a bit hard to follow.

R7: Thanks - we will revise this sentence to make it clear.

8. P5L18: why is the region vulnerable to climate change?

R8: Yes, we will add the explanation on this. For details, see reply R13 to Referee #1.

9. P6L3: it's not clear where the 2 WWTP's are located. Can you add them to your map?

R9: We will add the locations to the map in Fig. 1.

10. P6L8: how much are agriculture and WWTP's (and other sources) contributing in %?

R10: This comparative N budgeting is more part of the result and discussion sections. We will add the numbers in Section 3.2 referring to the current export situation. To give an idea: currently the fraction of wastewater at the total catchment nitrate export is 14%. Note that this fraction is removed from the exported nitrate in our analysis to focus on the diffuse pathways only (see P12, L9-27).

11. Figure 1: the stream is not very clear on this map. R11: We will highlight the river system.

12. P7L5 :"artificially drained" Do you mean drained by open ditches or by subsurface tube drains? How much has subsurface tube drainage?

R12: Yes, we will differentiate between "open ditches" and "tile drains" in the sentence by adding corresponding percentages. While more than half of the drains in the mid-stream sub-catchment are tube drains, the downstream sub-catchment is much more dominated by open ditches.

13. Table 1: The fraction artificially drained (last row) is much lower downstream. I

would expect more artificial drainage in the downstream part of the catchment as this is usually the wetter part of the catchment. Is there a reason why there is less artificial drainage needed in the downstream part?

R13: Thank you for this remark. We think this is related to the hydro-climatic conditions. The downstream area is significantly warmer and dryer in comparison to the colder and wetter upstream areas (see also response R20 for Referee #1 on this issue). This is also reflected in discharge behavior - the strong drop of discharge contribution from the different sub-catchments as indicated in Table 2.

14. P8L30 ": : :we do not account for wastewater fluxes at this point: : :" Why is this legitimate? Is the wastewater N flux negligible?

R14: We focused on diffuse N pathways via soil and groundwater where the legacy accumulation and time lags between input and output can potentially occur. Therefore we discounted the point contribution from both WWTPs from our N-data prior to TT analyses. See also the reply R10 above for the contribution of the WWTPs,

15. Figure 2 and 3: shouldn't these figures be presented in the results section?

R15: We understand your remark, but we still favor these figures related to data presentation in this section as it is now in the manuscript (see a similar example in Tetzlaff et al., (2014)). It's a presentation of the measured raw-data, while the results present the derived aggregated concentration and fluxes after using the WRTDS method. We will adjust the concerned section heading to "Data and methods" so to make it more clear.

16. Figure 2c: It seems like the NO3 concentration is 0 around 2007 and at the end of the graph. Please check this. There also seems to be a regime-shift in this plot just before 2000. What happened?

R16: You are right, we will correct these data points dropping to zero in Fig. 2 (see comment to that in response R15 of the Referee #1). The visible regime-shift around

2000 is related to the changing C-Q relations at the time where the dilution pattern switches to the enrichment pattern (see also Fig.7 c1 and c2). We address that in section 3.5 and in the discussion.

17. P11L6: "flow-normalized concentrations" It is not clear here why you need flow normalization. Consider to bring forward the end of the paragraph. Why would you want to take out the impact of variable flow conditions?

R17: We will drop the wording "flow-normalized" from here as the reasoning and procedure for the normalization is explained later on in this section (P12, L5-8).

18. P11L9: I don't understand how you interpolate the bi-weekly/monthly data. ": : :using a flexible statistical representation for every day of the discharge record".

R18: We will carefully revise that section to make methods more clear. The interpolation is based on a regression model using Q as a predictor, a trend component and a seasonal (sinusoidal) component. This model is fitted for every day separately utilizing a weighted regression approach that weights observation before and after that day differently based on their relevance for that specific day. Details are given in Hirsch et al. (2010). We note a mistake in the references here, and will correct this in the revised manuscript (the citation Hirsch & DeCicco in the text refers to the R-package while in the reference list the according paper Hirsch et al. is cited).

19. P13L14: "purple line"!purple dashed line

R19: Thanks - we will change that in the revised manuscript.

20. P13L21: "peaked 1980" ! peaked in 1980 R20: Thanks - we will change that in the revised manuscript.

20. Table 2: It is hard to connect the numbers for the LFS and HSF contributions in the text (<10%, 33%) with this table. It would be better not to give the cumulative contributions, so for HFS: 21, 69, 10.

R20: We will revise the table to avoid confusion between cumulative and single sub-catchment information.

21. P15L11: I don't understand ": : :besides the statistical evaluation of the time series"

R21: Thanks - we will revise this sentence to make it clear.

22: P16L6-15: During the measurement period the catchment will partly export N-inputs from before 1970/76. This could be seen as the legacy of the period before the measurement period. The missing N described here adds to the legacy from before 1970/76.

R22: Your point is right and we are aware of this discrepancy. We tried to underline this problem by stating: "overlapping time period of in- and output". A more appropriate comparison of in- and output would only be possible with the exact knowledge of TTs. In this first view of input-output-differences, we took the corresponding years for a quantitative comparison. Later on in the conclusions (P27, L22-25), we shift the input to the output ("assuming the temporal offset of peak TTs between in- and output of 12 a") and quantify the imbalance between both. With an additional sentence in the concerned section, we will underline this difference in a better way.

23. P18L6: why are these TTs for all seasons taken together not presented?

R23: We will add these lines in Fig. 6.

24. Figure7b1: The concentrations seem to drop here, before the input drops. How is this possible?

R24: Of course input changes cannot affect output earlier on. We think this drop in riverine nitrate concentration around 1985 is rather related to the sharp stop of increasing N-input at the beginning to mid of the 1970s and the following decrease of inputs.

25. Figuret7c1: The higher concentrations in summer and fall during the peak around

1990 are surprising. This would indicate that the concentrations in deep groundwater with long travel times to surface water are higher than the concentrations in shallow groundwater with short travel times. Is this groundwater N that infiltrated in the midstream catchment and seeps up in the downstream catchment?

R25: We can understand your confusion, but as explained in Section 4.2, the higher concentrations downstream in summer and fall are result of different nitrate source contribution during LFS and HFS. HFS-signals downstream are dominated by contributions from the wetter midstream sub-catchment with higher discharge per area and generally lower concentrations (see also Table 2 and similarity of midstream and downstream high flow concentrations shown in Fig. 7), whereas the low flow concentrations are dominated by the groundwater discharging from the downstream sub-catchment with much lower groundwater recharge and likely higher groundwater nitrate concentrations.

26. Figure 7a2-c2: add a legend. R26: We will add the color gradient to Fig. 7.

27. P20L9: refer to figure 7a2. R27: We will add the suggested reference.

28.P20L7-P21L17: This text in combination with figure 7 is quite a hard puzzle.

R28: Thank you for that comment. We will carefully revise this section and pay close attention to the information needed for the discussion later on.

29. P22L1: "was difficult" ! "was impossible" R29: Thanks, we will revise that sentence.

30. P22L1-2: Degradation of organic matter may play a role.

R30: You are right. We add this aspect in the sentence.

31. P22L17-20: I don't understand why "steeper terrain suggests a deeper infiltration" and "leaching of NO3 from a wider depth range than flat terrains". I would expect the opposite; deeper infiltration and leaching from a wider depth range in flat terrains. Of course, this depends on the geology.

R31: We would like to refer the paper by Jasechko et al. (2016) at this point: "Conversely, the reduced prevalence of young streamflow in steeper terrain suggests that steeper landscapes tend to favor deeper vertical infiltration rather than shallow lateral flow. A tendency for greater infiltration in mountainous watersheds may seem counterintuitive, but is consistent with conceptual models of runoff generation and groundwater flow that suggest that topographic roughness drives long groundwater flow pathways that bypass first-order streams."

32. P22L26:"to for an" ! "for an"

R32: Thanks, we will change this as suggested.

33. P23L9-10: "Hence,: : :output" I think that this conclusion that denitrification is weakly supported by the previous text. Groundwater quality measurements would be very useful here.

R33: We hope to improve the overall argumentation through a support by findings in another study Hannappel et al. (2018) who analyzed groundwater and an enhanced discussion of the isotope evidence by Müller et al. (2018) – see more information in response R4 for Referee #1.

34. P23L16: why did Kuhr et al exclude denitrification?

R34: We would drop this "grey" citation (this is a report, not peer reviewed) in our manuscript and refer to Hannappel et al (2018) – see the previous comment R33.

35. P24L4-9: from this paragraph and especially the last 2 sentences it seems like it is not important whether the legacy store is growing or the denitrification capacity is used, however on P22L23-25 you stated that this difference is important.

R35: We will make our point clearer in the revised text. With the long-term data collection, we can only hypothesize whether the missing N is stored or denitrified, although it would be important for management. Beside management advices, we can show that the catchment N-input is unsustainable high, either due to the ongoing build-up of an

even bigger legacy or due to relying on a denitrification capacity which is unlikely to be infinite.

36. Figure 8: This figure does not make any sense to me.

R36: We will revise this figure to make our conceptual understanding of the catchment more clear.

37. P25L3-5: I don't think that you can make this assumption; the flow contributions from a certain depth can vary a lot due to interannual variability

R37: We don't think that there is evidence of a long-term change of flow paths in the catchment. Hydroclimatic conditions did not change; land use, topography and river network are stable over the long observation period. We will add these aspects here to better justify our assumption.

38. P26L3: You can also argue that groundwater seeping up is more important in the downstream catchment. This would mean more discharge of relatively old water.

R38: The TTs in the downstream part are shorter than those in Midstream, and not the other way round. Our argumentation is based on the greater prevalence of young streamflow in flatter terrain as shown also by Jasechko et al. (2016). See also response R31.

References uses (that are not in the main manuscript) • Hannappel, S., Kopp, C. and Bach, T. (2018) Characterization of the denitrification potential of aquifers in Saxony-Anhalt. Grundwasser 23(4), 311-321. • Tetzlaff, D., Birkel, C., Dick, J., Geris, J. and Soulsby, C. (2014) Storage dynamics in hydropedological units control hillslope connectivity, runoff generation, and the evolution of catchment transit time distributions. Water Resources Research 50(2), 969-985.

---

## Author Response (AR1)

**The following document contains the responses to the reviewers, the manuscript with tracked changes and the revised manuscript with comments connecting the changed parts to the points raised by the reviewers**

**Response to anonymous Referee #1**

General response: We thank the referee for the valuable inputs and remarks. We address all comments below and hope to clarify the questions raised. In the manuscript with tracked changes notation on the responses can be found.

**1. How is error propagated? The authors often report four significant figures, but do not report standard deviation, confidence intervals, or some other estimate of uncertainty. Given the compound assumptions of the input chronicle models and the hydrological components, a sensitivity analysis or some kind of quantification of uncertainty seems warranted.**

R1: Right, an uncertainty analysis was missing so far and will for sure improve the analysis. We derived the confidence band of flow normalized concentration and fluxes based on a bootstrap method for WRTDS proposed by Hirsch et al. (2015) and available in the egretCI package in the R environment. We estimated $5^{th}$ and $95^{th}$ percentiles of flow normalized concentration and fluxes for each year of measurements in a conservative best case/ worst case analysis in the input-output budgeting and in the estimation of travel times. We added a section in the methodology, describing this and changed table 3, and 4 and figures 3, and 4 respectively. For the input of nitrogen (nitrogen surplus) we stated the methodological error provided by Bach & Frede (1998) in the method section as well.

**2. The idea of comparing biogeochemical and hydrological legacies is very compelling but it remains unclear to me how these parameters were estimated and compared. Structuring the methods around the research questions or overarching hypotheses and carrying this through the manuscript would make this flow clearer would make the results/discussion more impactful.**

R2: That is a very helpful comment. We revisited the introduction and the research questions and made them more clear especially on where we see the potential of using the C-Q relationships to better disentangle the biogeochemical and hydrological legacies. Moreover, we wrote introductory sentences for the different method parts to better integrate research questions with the method steps. Finally we made the discussion on this topic more explicit in the discussion part as well. Based on the results of our analyses, we improved upon the discussion part raising new hypothesis on dominant legacy types. Since our study is based on a data-driven analysis, we can't test such new hypothesis – but certainly we feel it is worth raising them from the data evidence so that a future initative could start looking into those new aspects.

**3. I think the discussion would be more engaging if the authors focused on the applicability of this approach to catchments generally, rather than explaining specific observations from their study. They do this effectively several times (e.g. starting on page 22 starting around line 20), but there is also quite a bit of retreatment of the results, which are specific to these sites.**

R3: We carefully reviewed and revised the sections that are specific to our catchments - shortening them without losing the main message. Here and in the conclusions we more explicitly indicated where we discuss and draw conclusions for the studied catchment and where we can generalize our findings. We also see a greater potential of applying this local

analysis to a wide range of catchments where we can more easily draw general conclusions (Page 29 Line 30 ff.).

**4. The authors present an interesting puzzle of massive nitrogen retention/removal that cannot be attributed to typical pathways (e.g. denitrification, uptake, mineral association). The authors then conclude that N storage (the biogeochemical and hydrological legacies) account for the disconnect. However, the dismissal of denitrification seems to be based on a few studies from this area, which are not described in detail (e.g. Page 23, line 15). If these other studies are definitive and reliable, more description of their methods should be given. Another explanation is associated with point 1 - could the N removal be much lower when uncertainty in inputs and outputs are included?**

R4: Yes – as mentioned above in R1 we addressed the uncertainty of the regression approach and the N input from agricultural areas. We would like to note here a recent paper published in November 2018 giving an overview on denitrification potential in the federal state this catchment is part of (Hannappel et al., 2018). It connects hydrochemical analysis of groundwater nitrate, oxygen and redox potential to the hydrogeological units in this region and states a general weak potential for denitrification for the study site. We included this study with methodological details to strengthen our argumentation on the denitrification part. We already included the study by Müller et al. (2018) within the same study area that provided strong evidence on the lack of denitrification based on their assessment on isotopic signatures in the integrated nitrate signal in the surface water. We put more emphasis on discussing this study as well to better argue our case.

**5. Line edits Page 2 Line 5: (Elser et al. 2007)**

R5: We added this reference here.

**6. Line 6: It seems odd to say these changes were strictly terrestrial. It seems they influenced both.**

R6: We dropped the word "terrestrial" at the specified location.

**7. Line 10: Do the authors mean the natural rate of reactive N fixation has been doubled (e.g. (Vitousek et al. 1997))?**

R7: Yes, Vitousek et al. (1997) and Smil (1999) refer to the same: Human activities are mainly responsible for doubling the amount of reactive/ biological active N that enters the element's cycle from the unreactive atmospheric pool of $N_2$. We added this reference and adjusted the sentence, accordingly.

**8. Page 3 Line 2: management interventions (instead of "measures")?**

R8: Thanks – we changed that.

**9. Line 2: Recent study from similar agricultural and climatic context that found decadal hydrologic (Kolbe et al. 2016; Marçais et al. 2018)**

R9: Thanks for the suggestion. We, refer to time lags of nitrate in response to interventions in the catchment here. The suggested studies address water travel time without making the connection to time lags in nitrate are therefore not eligible here but are used later on in the manuscript.

**10. Line 16: I actually think there are quite a few studies, especially recently (Dupas et al. n.d.; Howden et al. 2010; Burt et al. 2011; Minaudo et al. 2015; Meter & Basu 2017; Abbott et al. 2018; Coble et al. 2018; Garnier et al. 2018; Marcé et al. 2018; Pinay et al. 2018; Fanelli et al. 2019)**

R10: Thanks for the input. We adjusted the sentences adding four of the suggested studies.

**11. Line 20: How do these analyses compare with soil-surface N balance approaches that include a crop and livestock removal component (Poisvert et al. 2017; Abbott et al. 2018)?**

R11: Both, Jawitz and Mitchell (2011) as well as Musolff et al. (2015) are not based on N balances but on an interpretation of the temporal dynamic (or lack of temporal dynamic) in the observed nitrate concentrations. We added that to this sentence. Our paper aims at a combination of both approaches – N balancing (since the N-input takes crops and livestock into account) and C-Q assessment. Both, Poisvert and Abbott refer to a comparable data basis for N-surplus as we do.

**12. Line 30: Recent paper on concentration-discharge responses to catchment saturation (Moatar et al. 2017)**

R12: Moatar et al. (2017) do not state what we wanted to say here for nitrate – an increase of "chemostasis" with increasing intensification of agriculture. We therefore did not include this citation at this point in the manuscript.

**13. Page 5 Line 18: In what dimensions is this catchment especially vulnerable to climate change?**

R13: A recent study by Wollschläger et al (2017) states a high vulnerability due to low water availability and a pronounced risk of summer droughts that is likely to be exacerbated by decreasing summer precipitation and increasing temperature/ potential evapotranspiration. One new reference stating that were included here as well (Samaniego et al. 2018). We added this information in the revised manuscript.

**14. Page 8 Line 13-20: Interesting that the primary datasets do not include non-agricultural land for N deposition. Why did the authors not use one of the products that provided a consistent N deposition rate across land-use types? Perhaps this is a small portion of the overall N budget, but it would be worthwhile to specify.**

R14: We combined two products for N input to agricultural and non-agricultural land as there is no consistent product available in Germany, covering both with the required spatial and temporal resolutions. We added this information to the text.

**15. Page 9**
**Figure 2: The dissimilarity in the NO3 concentration time series is striking as are the drops to zero mg/L even at the lowest site. Consider combining Figures 2 and 3 to allow visual comparison of discharge and concentration.**

R15: The "drops to zero" are actually the no-data-values that are erroneously displayed as zero (but not considered in the WRTDS regressions). We adjusted the figure to properly reflect the missing information; and also combined Fig. 2 with the discharge in Fig. 3.

**16. Page 10**
**Line 9: the discharge time series were used. . .**

R16: Thanks – we changed this in the revised manuscript.

**17. Page 11**
**Line 8: allows increasing . . .**
R17: Thanks – we changed this in the revised manuscript.

**18. Page 12**
**Line 10: Because our purpose was to balance and compare . . .**
R18: Thanks – we changed this in the revised manuscript.

**19. Line 12: This justification seems unclear. Is it simply claiming that the longer-term trends are accurate, though the daily values are not?**

R19: No, the daily values are accurate but just that they not available at a daily time scale. We thus refer to the robust aggregated annual wastewater flux that much better fits to the flow normalized fluxes provided by the WRTDS regression analysis (see statement P11L31ff). Daily values are used to estimate an average fraction of $NO_3$-N in the wastewater N flux.

**20. Page 14**
**Table 2: These differences in specific discharge are remarkable. Is this typical for this area or is the three-fold difference due to a known environmental or anthropogenic variable?**

R20: Yes this is remarkable but typical, and one of the reasons behind the establishment of the TERENO observatory system (Wollschläger et al. 2017). Wollschläger et al. (2017) state the strong precipitation gradient from 1700 mm/ a down to less than 500 mm/a within a range of 50 km due to the rain shadow of the Harz mountains; and thereby leading to strong spatial differences in the resulting specific discharges. We made this fact more clear in the method section.

**21. Page 15**
**Line 11: Revise sentence for grammar and clarity (with implications for instead of with discussion on?)**

R21: Thanks - we revised the sentence.

**22. Page 16**
**Line 14: It is striking that the retention capacity increases 5-fold with landscape position. Is this because of shifts in soil and subsurface properties or because the retention or removal rates are dependent on substrate concentration?**

R22: Yes, this is quite a strong difference that is stated here as an observed result. Discussion on the reasoning can be found later on in Section 4.1.

**23. Page 22**
**Line 20: Nitrification also results in gaseous N loss via the "leaky pipe" pathway (Hart et al. 1994).**

R23: Right – there can be losses of $N_2O$ leaving the system at the nitrification step. However, in comparison to denitrification it does not appear to be a dominant loss term in N-budgets (Rivett et al. 2008, Galloway et al. 2004). See also comment R4 – the paper by Müller et al. (2018) on the isotope evidence for the lack of N removal in the study catchment.

**24. Line 29: Is this referring to denitrification in the near-surface zone or throughout the whole catchment? With pyrite, sulfur, and other iron ubiquitous in the weathered and fractured zones, aquifer denitrification is likely occurring**

R24: We refer to denitrification in general, taking both autotrophic and heterotrophic denitrification into account. Both need the absence of oxygen independent of whether electron donors are available or not. Also both affect the finally measured isotope signature in the remaining nitrate in the stream. See also our comment R4 with the new study (Hannappel et al. 2018) stating the lack of denitrification evidence that we included in the revised manuscript.

**25. Page 23 Line 18: New methods for constraining aquifer travel time to constrain removal rates using numerical or empirical methods (Kolbe et al. 2016; Marçais et al. 2018).**

R25: Right. Enhanced knowledge on water travel time will improve the estimation of reaction rates. We considered Marcais et al (2018) and the more recent study by Kolbe et al. (2019) in the conclusion of the revised manuscript.

**26. Page 25 Line 1: Similar to these observations, though they are on a much smaller scale (Thomas & Abbott 2018)**

R26: Thanks – we considered this in the revised manuscript.

**27. Page 28 Line 9: were explained**
R27: Thanks – we changed that as suggested.

**28. Line 14: catchment reaction seems like an odd description for transit time.**

R28: That is right. We changed that phrase as suggested.

**Response to anonymous Referee #2**
General response: We thank the referee for the valuable inputs and remarks. We address all comments below and hope to clarify the questions raised. In the manuscript with tracked changes notation on the responses can be found.

**General comments:**
**1. The manuscript addresses the important issue of legacy stores of nutrients, which may prevent mitigation actions that reduce the inputs from having immediate effects on stream water quality. I like the date drive approach to investigate the travel times of nitrate. The paper shows that 85% of the N input is retained within the catchment. The investigation about the fate of this lost N is not very convincing and inconclusive. Based on data on inputs and outputs alone, the authors cannot proof whether the N is retained in the soil, whether it is traveling along long flow paths, or whether it is denitrified. The authors try to give answers based on literature, but this is not very convincing. A weak point of the paper is that the entire soil and groundwater system is addressed as a black box. This is a bit strange given the focus on the paper on N – stores and travel times in soil and groundwater. Including data on e.g. groundwater heads and flow-paths and concentration depth profiles for N could provide more certainty about the fate of the lost N.**

R1: We understand the reviewer remark. With the current datasets at hand and rather invoking any model conceptualization, our study can only hypothesize and argue about travel times (TTs) and legacy, which is unfortunately a common methodological challenge of studies on catchment scale. We treat the entire catchment including soil and groundwater system as a black box and try to understand the inherent processes by looking at the signals produced or altered by this box. By doing a data-driven analysis, our aim is to provide observation-based evidence on the system input-output response behavior, which can then be a starting point for developing either more targeted field-based or model-based "mechanistic" studies. Groundwater measurements and soil profiles would be a great help to support the hypothesis, but those observational records are generally not available. We tried hard to overcome this lack of knowledge by strengthening with the isotopic evidence on a minor role of denitrification, by incorporating a new regional study on denitrification, and by a comprehensive literature review for our studied catchment and comparable study sites (see also response R4 for Referee #1).

**Specific comments:**
**2. Title: Consider to leave out 'decadal'. I don't understand why you would only be looking at decadal trajectories**

R2: Agree, we dropped "decadal" in the title.

**3. Abstract: The abstract is rather long. Especially the description of the results (from "We show: : :"). Consider to start a new paragraph here to make the structure more clear. The conclusion statement is a bit weak. Management should both address longer term and short term N-loads. How does this change water quality management in practice?**

R3: We carefully reviewed the abstract, shortened the result section and put a clear focus on consequences for management practice that results from our analysis.

**4. From P3L4 until P4L22 the introduction reads like a description and a justification of the methods that you apply. It remains unclear what is not yet known from the existing scientific literature, why that is important, and what new science this paper brings.**

R4: We carefully reviewed the Introduction part of the paper to highlight the suggested mentioned aspects: What is known? Why it is important? And what's new in our study? This is also in line with comment 2 of the Reviewer 1. We made the importance and the new scientific messages conveyed from our work clearer in the revised version.

**5. In P4L20 you state that "data-driven studies focused either solely on N-budgeting and legacy estimation or on TTs." What data-driven studies do you mean here? Why is this a problem / what problem do you solve by combining these? The referencing to Van Meter and Basu is quite excessive.**

R5: We refer to the data-driven studies e.g. by Worral et al., 2015 and Dupas et al., 2016. We have adapted the introduction text to further underline the advantage of combining the quantification of legacy and TT in one study (and from the same database) to use the TTs to explain the legacy. In terms of studies cited: See also comment R10 for Referee #1 where we aim at including a greater variety of studies.

**6. P2L11-12: here you state that the agricultural nitrogen input is still high since the 1980's. It did decrease in most EU member states since the '80s as a result of the introduction of manure legislation, didn't it?**

R6: The reviewer is right in pointing this out that N-inputs, also from agricultural sources were reduced (but they are still on a high level). We rewrote the concerned sentence to correct this inaccuracy.

**7. P2L26: "The evaluation of measures: : :" What evaluation of measures. This sentence is a bit hard to follow.**

R7: Thanks - we revised this sentence to make it clear

**8. P5L18: why is the region vulnerable to climate change?**

R8: Yes, we added the explanation on this. For details, see reply R13 to Referee #1

**9. P6L3: it's not clear where the 2 WWTP's are located. Can you add them to your map?**

R9: We added the locations to the map in Fig. 1.

**10. P6L8: how much are agriculture and WWTP's (and other sources) contributing in %?**

R10: Referring to the last 5 years of observations, $NO_3$-N load from wastewater made up 17% of the total observed $NO_3$-N flux at the midstream station (see below) and 11% at the downstream station. We added this information here. Note that this fraction is removed from the exported nitrate in our analysis to focus on the diffuse pathways only (see P11, L29ff).

**11. Figure 1: the stream is not very clear on this map.**
R11: We highlighted the river system.

**12. P7L5 :"artificially drained" Do you mean drained by open ditches or by subsurface tube drains? How much has subsurface tube drainage?**

R12: We now differentiate between "open ditches" and "tile drains" in the sentence by adding corresponding percentages. While more than half of the drains in the midstream subcatchment are tube drains, the downstream sub-catchment is much more dominated by open ditches.

**13. Table 1: The fraction artificially drained (last row) is much lower downstream. I would expect more artificial drainage in the downstream part of the catchment as this is usually the wetter part of the catchment. Is there a reason why there is less artificial drainage needed in the downstream part?**

R13: Thank you for this remark. This is related to the hydro-climatic conditions. The downstream area is significantly warmer and dryer in comparison to the colder and wetter upstream areas (see also response R20 for Referee #1 on this issue). This is also reflected in precipitation and discharge behavior – particularly in the strong drop of discharge contribution is noted from the different sub-catchments as indicated in Table 2.

**14. P8L30 ": : :we do not account for wastewater fluxes at this point: : :" Why is this legitimate? Is the wastewater N flux negligible?**

R14: We focused on diffuse N pathways via soil and groundwater where the legacy accumulation and time lags between input and output can potentially occur. Therefore we discounted the point contribution from both WWTPs from our N-data prior to TT analyses. See also the reply R10 above for the contribution of the WWTPs.

**15. Figure 2 and 3: shouldn't these figures be presented in the results section?**

R15: We understand your remark, but we still favor these figures related to data presentation in this section as it is now in the manuscript (see a similar example in Tetzlaff et al. (2014)). It's a presentation of the measured raw-data, while the results present the derived aggregated concentration and fluxes after using the WRTDS method. We adjusted the concerned section heading to "Data and methods" so to make this clearer.

**16. Figure 2c: It seems like the NO3 concentration is 0 around 2007 and at the end of the graph. Please check this. There also seems to be a regime-shift in this plot just before 2000. What happened?**

R16: You are right, we corrected these data points dropping to zero in Fig. 2 (see comment to that in response R15 of the Referee #1). The visible regime-shift around 2000 is related to the changing C-Q relations at the time where the dilution pattern switches to the enrichment pattern (see also Fig.7 c1 and c2). We address that in section 3.5 and in the discussion.

**17. P11L6: "flow-normalized concentrations" It is not clear here why you need flow normalization. Consider to bring forward the end of the paragraph. Why would you want to take out the impact of variable flow conditions?**

R17: We dropped the wording "flow-normalized" from here as the reasoning and procedure for the normalization is explained later on in this section (P12, L5-8).

**18. P11L9: I don't understand how you interpolate the bi-weekly/monthly data. ": : :using a flexible statistical representation for every day of the discharge record".**

R18: We carefully revised that section to make methods more clear. The interpolation is based on a regression model using discharge (Q) as a predictor, a trend component and a seasonal (sinusoidal) component. This model is fitted for every day separately utilizing a weighted regression approach that weights observation before and after that day differently based on their relevance for that specific day. Details are given in Hirsch et al. (2010). We noted a mistake in the references here, and corrected this in the revised manuscript (the

citation Hirsch & DeCicco in the text refers to the R-package while in the reference list the according paper Hirsch et al. is cited).

**19. P13L14: "purple line"!purple dashed line**

R19: Thanks - we changed that in the revised manuscript.

**20. P13L21: "peaked 1980" ! peaked in 1980**
R20: Thanks - we changed that in the revised manuscript.

**20. Table 2: It is hard to connect the numbers for the LFS and HSF contributions in the text (<10%, 33%) with this table. It would be better not to give the cumulative contributions, so for HFS: 21, 69, 10.**

R20: We revised the table to avoid confusion between cumulative and single sub-catchment information.

**21. P15L11: I don't understand ": : :besides the statistical evaluation of the time series"**

R21: Thanks - we revised this sentence to make it clear.

**22: P16L6-15: During the measurement period the catchment will partly export N-inputs from before 1970/76. This could be seen as the legacy of the period before the measurement period. The missing N described here adds to the legacy from before 1970/76.**

R22: Your point is right and we are aware of this discrepancy. We tried to underline this problem by stating: "*overlapping time period of in- and output*". A more appropriate comparison of in- and output would only be possible with the exact knowledge of TTs. In this first view of input-output-differences, we took the corresponding years for a quantitative comparison. Later on in the conclusions (P27, L22-25), we shift the input to the output ("*assuming the temporal offset of peak TTs between in- and output of 12 a*") and quantify the imbalance between both. We added a sentence in the concerned section, to underline this difference in a better way.

**23. P18L6: why are these TTs for all seasons taken together not presented?**

R23: We added these lines in Fig. 5.

**24. Figure7b1: The concentrations seem to drop here, before the input drops. How is this possible?**

R24: Of course input changes cannot affect output earlier on. We think this drop in riverine nitrate concentration around 1985 is rather related to the sharp stop of increasing N-input at the beginning to mid of the 1970s and the following decrease of inputs.

**25. Figuret7c1: The higher concentrations in summer and fall during the peak around 1990 are surprising. This would indicate that the concentrations in deep groundwater with long travel times to surface water are higher than the concentrations in shallow groundwater with short travel times. Is this groundwater N that infiltrated in the midstream catchment and seeps up in the downstream catchment?**

R25: We can understand your reasoning, but as explained in Section 4.2, the higher concentrations downstream in summer and fall are result of different nitrate source

contribution during low flow seasons (LFS) and high flow seasons (HFS). HFS-signals downstream are dominated by contributions from the wetter midstream sub-catchment with higher discharge per area and generally lower concentrations (see also Table 2 and similarity of midstream and downstream high flow concentrations shown in Fig. 6), whereas the low flow concentrations are dominated by the groundwater discharging from the downstream sub-catchment with much lower groundwater recharge and likely higher groundwater nitrate concentrations.

**26. Figure 7a2-c2: add a legend.**
R26: We added the color gradient to Fig. 6.

**27. P20L9: refer to figure 7a2.**
R27: We added the suggested reference.

**28.P20L7-P21L17: This text in combination with figure 7 is quite a hard puzzle.**

R28: Thank you for that comment. We carefully revised this section and payed close attention to focus on the information needed for the discussion later on.

**29. P22L1: "was difficult" ! "was impossible"**
R29: Thanks, we revised that sentence.

**30. P22L1-2: Degradation of organic matter may play a role.**

R30: Yes, NO3 may be released from organic matter. However, on the longer term there cannot be more release than input. As the balance indicate more export than import we rather think we either have unaccounted sources or overrestimated the biological N-fixation (underestimation of resulting N surplus). Both arguments are in the text.

**31. P22L17-20: I don't understand why "steeper terrain suggests a deeper infiltration" and "leaching of NO3 from a wider depth range than flat terrains". I would expect the opposite; deeper infiltration and leaching from a wider depth range in flat terrains. Of course, this depends on the geology.**

R31: For this point, we would like to refer the reasoning provided in the recent paper by Jasechko et al. (2016):
*"Conversely, the reduced prevalence of young streamflow in steeper terrain suggests that steeper landscapes tend to favor deeper vertical infiltration rather than shallow lateral flow. A tendency for greater infiltration in mountainous watersheds **may seem counterintuitive**, but is consistent with conceptual models of runoff generation and groundwater flow that suggest that topographic roughness drives long groundwater flow pathways that bypass first-order streams."*

**32. P22L26:"to for an" ! "for an"**

R32: Thanks, we changed this as suggested.

**33. P23L9-10: "Hence,: : :output" I think that this conclusion that denitrification is weakly supported by the previous text. Groundwater quality measurements would be very useful here.**

R33: We improved the overall argumentation made here through a support by findings in another study by Hannappel et al. (2018) who analyzed groundwater and an enhanced discussion of the isotope evidence by Müller et al. (2018) – see more information in response R4 for Referee #1.

**34. P23L16: why did Kuhr et al exclude denitrification?**

R34: We dropped this citation at this point and refer to Hannappel et al (2018) – see the previous comment R33.

**35. P24L4-9: from this paragraph and especially the last 2 sentences it seems like it is not important whether the legacy store is growing or the denitrification capacity is used, however on P22L23-25 you stated that this difference is important.**

R35: We changed this paragraph to make the point clearer in the revised text. With the long-term data collection, we can only hypothesize whether the missing N is stored or denitrified, although it would be important for management. Beside management advices, we can show that the catchment N-input is unsustainable high, either due to the ongoing build-up of an even bigger legacy or due to relying on a denitrification capacity which is unlikely to be infinite.

**36. Figure 8: This figure does not make any sense to me.**

R36: We have revised this figure to make our conceptual understanding of N-storages and release in the study catchment more clear.

**37. P25L3-5: I don't think that you can make this assumption; the flow contributions from a certain depth can vary a lot due to interannual variability**

R37: We don't think that there is evidence of a long-term change of flow paths in the catchment. Hydroclimatic conditions did not change; land use, topography and river network are stable over the long observation period. We added these aspects here to better justify our assumption.

**38. P26L3: You can also argue that groundwater seeping up is more important in the downstream catchment. This would mean more discharge of relatively old water.**

R38: The TTs in the downstream part are shorter than those in Midstream, and not the other way round. Our argumentation is based on the greater prevalence of young streamflow in flatter terrain as shown also by Jasechko et al. (2016). See also response R31.

References uses (that are not in the main manuscript)
- Tetzlaff, D., Birkel, C., Dick, J., Geris, J. and Soulsby, C. (2014) Storage dynamics in hydropedological units control hillslope connectivity, runoff generation, and the evolution of catchment transit time distributions. Water Resources Research 50(2), 969-985.

[revised manuscript text omitted]

Kommentar [SE20]: Ref1, No.14
Kommentar [SE21]: Ref1, No.2
Kommentar [SE22]: Ref1, No.14
Kommentar [AM23]: Ref1, No. 1

each subcatchment, was added to the agricultural N-surplus to achieve the total N-input per area and subcatchment. In contrast to the widely applied term net anthropogenic nitrogen input (NANI), we do not account for wastewater fluxes at this pointin the N-input but rather focus on the diffuse atmospheric deposition, biological N-fixation and agricultural input, N pathwaysN-input and connected flow paths where legacy accumulation and time lags between in- and output potentially occur.

**2.3 Nitrogen output**

**2.3.1 Discharge and water quality time series**

Discharge and water quality observations were used to quantify the N load and to characterize the trajectory of $NO_3$-N nitrate concentrations and the C-QC–Q trajectories in the three sub-catchments.

The data for water quality (biweekly to monthly) and discharge (daily) from 1970 to 2016 were provided by the LHW, Saxony-Anhalt.

The biweekly to monthly sampling was done at gauging stations defining the three subcatchmentsub-catchments (NO₃-N: Fig. 2; NH₄-N: S1.2.1; NO₂-N: S1.2.2). The data sets cover a wide range of in-stream chemical constituents including major ions, alkalinity, nutrients and in-situ parameters. As this study only focuses on N-species, we restricted the selection of parameters to nitrate ($NO_3$; Fig. 2), nitrite ($NO_2$; supplement S1.2.2) and ammonium ($NH_4$; supplement S1.2.1).

[Figure]

[Figure]

**Figure 222: NO₃-N concentration _and discharge (Q)_ time series: Upstream (a), Midstream (b) and Downstream (c).**

Discharge time -series at daily time scales were measured at two of the water quality stations (Upstream, Downstream; Fig. . 23). Continuous daily discharge series are required to calculate flow-normalized concentrations (Ssee the following Ssection 2.3.2 for more details). To derive the discharge data for the midstream station and to fill measurement gaps at the other stations (2-% Upstream, 3-% Downstream), we used simulations from a grid-based distributed mesoscale hydrological model . called mHM (Samaniego et al., 2010; Kumar et al., 2013). Daily mean discharge was simulated for the same time frame as the available measured data. We used a model set-up similar to Müller et al. (2016) with robust results capturing the

observed variability of discharge in the studied, near-by catchments. We note that the discharge time series  were used as weighting factors in the later analysis of flow-normalized concentrations. Consequently it is more important to capture the temporal dynamics than the absolute values. Nonetheless, we performed a simple bias correction method by applying the regression equation of simulated and measured values to reduce the simulated bias of modelled discharge. After this revision, the simulated discharges could be used to fill the gaps of measured data. The midstream station (Derenburg) for the water quality data is 5.6 km upstream of the next gauging station. Therefore, the nearest station (Mahndorf) with simulated and measured discharge data was used to derive  the bias correction equation that was subsequently applied to correct the simulated discharge data at the Midstream station, assuming the same bias between modelled and  observed discharges  at both near-by gauging stations.

**Kommentar [SE25]:** Ref1 Nr.16

[Figure]

Figure : Discharge time series: Upstream (a), Midstream (b) and Downstream (c).

**2.3.2 Weighted regression on time, discharge and season (WRTDS) and waste water correction**

The software package "Exploration and Graphics for RivEr Trends" (EGRET) in the R software environment by Hirsch and De-Cicco et al. (20109) was used to derive estimate daily flow normalized concentrations of $NO_3$-N . This tool enables an analysis, based on the long term changes in water quality and streamflow, using the water quality method utilizing a "Weighted Regressions on Time, Discharge, and Season" (WRTDS; Hirsch & De Cicco, 2010 Hirsch et al. 2010). The WRTDS method allows the interpolation of irregularly sampled to increasing the temporal resolution of concentration

Kommentar [SE26]: Ref2 No. 17

Kommentar [SE27]: Ref1 No. 17

[revised manuscript text omitted]

**Kommentar [AM36]:** Ref. 1, No. 1

Calculated loads (Fig. 3<s>4</s>b) also showed a drastic change between the beginning and the end of the time<s>-</s>series. The daily upstream load contribution was below 10<s>-</s>% of the total annual export at the downstream station in all decades and then the estimates decreased from 9<s>-</s>% (1970s) to 4<s>-</s>% (2010s). The median daily load between 1970s and 1990s tripled midstream (0.1 t d$^{-1}$ to 0.3 t d$^{-1}$) and more than doubled downstream (0.2 t d$^{-1}$ to 0.5 t d$^{-1}$). In the 1990s, the Holtemme River exported on average more than 0.5 t d$^{-1}$ of NO$_3$-N, which, related to the agricultural area in the catchment, translates into more than 3.1 kg N d$^{-1}$ km$^{-2}$ (maximum of <s>-</s>13.4 kg N ha$^{-1}$ a$^{-1}$ in<s>,</s> 1995).

**3.3 Input-Output-balance: N-budget**

<s>Besides the statistical evaluation of the time series itself, the</s>We jointly evaluated the estimated N-inputs <s>were associated</s><s>with</s>and the exported NO$_3$-N loads to enable an input-output-<s>assessment</s>balance. <s>The estimated N-inputs were associated with the exported loads of the</s> <s>subcatchment</s>sub-catchment <s>besides the statistical evaluation of the time series.</s> This <s>connection</s> comparison on the one hand allowed for an estimation of the catchment's retention potential <s>with a discussion on potentially accumulated biogeochemical and hydrological legacy</s>, and on the other hand <s>it</s> enabled us to <s>predict</s> estimate future exportable loads.

**Kommentar [SE37]:** Ref.2, No.21

**Kommentar [SE38]:** Ref1, No.21

[revised manuscript text omitted]

---

## Author Response (AR2)

**General comment:**

We thank both referees for the valuable remarks and feedback.
In the process of revising the paper, we unfortunately came across a calculation error regarding $N_{input}$ via biological fixation. As it was described in the methods previously, we erroneously considered this fixation as a sink (in forests and grasslands) and not as a source of N in the catchment.
This resulted in an under-estimation of N inputs from non-agricultural sources that is especially for the up- and midstream sub-catchments important (Midstream: 57 %, Downstream: 36 %). The estimated N input increased from 42 758 t to 53 437 t (1976–2015) and the corresponding retention rates changed, especially in the midstream catchment with a higher fraction of non-agricultural land (from 54 % to 75 %). Also the derived travel times changed marginally, due to the modified N input. Although atmospheric deposition and biological fixation (constant over time as land use change is negligible) increased the absolute N input, the agricultural input with its historical dynamics is still the dominant N source and driver for our investigated processes. Note that our statements and conclusions were not affected by this adjustment. We apologize for this mistake!

**Suggestions for revision or reasons for rejection (will be published if the paper is accepted for final publication)**

The authors managed to clearly improve the manuscript based on the first reviews. I only have a few more suggestions:

1. P2 L27-29: In many cases, surface water quality does improve quickly after reducing the N inputs. See for example Rozemeijer et al., 2014
Rozemeijer, J.C., J Klein, HP Broers, TP van Tol-Leenders, B Van Der Grift, 2014. Water quality status and trends in agriculture-dominated headwaters; a national monitoring network for assessing the effectiveness of national and European manure legislation. Environmental monitoring and assessment 186 (12), 8981-8995.

Yes - we revised this sentence and incorporate your suggested citation showing that both, long time lags and fast recovery have been reported.

2. P4 L5-6: Chemostatic behavior after decades of N inputs is not what we've observed in the Netherlands. We still see changing concentrations during events. Quite often, the concentration dips shortly during events due to fast runoff of water that did not have time to take up nitrate from the subsurface. But also: unless decades of N inputs, we see decreasing groundwater concentrations with depth, due to denitrification. This still leads to higher concentrations during the wet season. For chemostatic behavior you would need similar concentrations in all flow routes from the deep groundwater to the direct runoff. We haven't seen that yet.

Thanks - we revised this sentence and made clear that effective denitrification can result in chemodynamic exports even with intensive agriculture.

3. P8 L7: What crops are grown and what type of fertilizer is used?

We add information on crop rotation from a study in the neighboring Selke catchment with comparable conditions that is also part of the TERENO project. We could not find explicit information on the type of fertilizers used in our catchment.

4. P9L29: What are 'in situ parameters'?

We added details on these parameters.

5. P11 L11-18: Do you have information about the performance of this method for interpolating concentration data? Is it better than a more straightforward linear or stepwise interpolation? And why is it better?

Our approach was, to detect system changes of the watershed caused by long-term human N-input. Discharge-driven variability makes the identification of trends difficult, because of the impact of year-to-year discharge variation. Therefore, the WRTDS-Method with its flow-normalization was used to get an understanding of "how concentration changes over time histories". This method covers discharge, seasonality and time dependence, which is a big advantage in comparison to solely linear interpolation of two parameters. An estimation point of a "long distance" to the three dimensions (time, season, discharge) will diminish its importance for the regression. The product of the three individual weights determines the overall weight of the data point (Hirsch et al., 2010). The bilinear flexible interpolation in response to time and ln(Q) overcomes limitation of a simple linear interpolation and, moreover, allows for long-term changes in the C–Q relation. This change in the export regime was one of our hypotheses (chemodynamic vs. chemostatic) and already known from other catchments in Europe and the US (Thompson et al., 2011; Dupas et al., 2016). Therefore a simple linear interpolation may ignore pattern in the data and would likely miss concentration peaks/ drops under event flow conditions. Furthermore, the method is implemented in the R-package "Exploration and Graphics for RivEr Trends (EGRET)" that is widely applied in comparable studies (van Meter et al., 2017; van Meter & Basu, 2017; Sprague et al., 2011).

6. P13 L 6: 97% of agricultural N is quite a lot given that the input from atmospheric deposition usually is in the range of 20% (L17-18). What is the remaining 3%?
Mention again here that you left out the effluent from the sewage treatment plants.

Yes, the reviewer was right. On a careful review of the calculations for N-input, we had to fix an error and had to revise several values in the paper. Despite these corrections, no content-related statements changed. The percentage N from of atmos. deposition decreases from 54% upstream to 5% downstream (midstream 34%).

7. P15 Figure 3: Consider to add some more detailed information about the NO3 concentrations in time. It would be good to see the year-to-year variations and the seasonal variations as well.

Here the text mainly refers to decadal changes. Year to year variations and seasonal changes can be found in Fig. 6. We added a reference to that figure earlier in the text.

8. P16 L6: You only use the export via surface water. Could there be export via (regional) groundwater as well? Why is it legitimate to neglect this?

We will adjust the wording to underline our focus on riverine load.

For regional groundwater flow out of the catchment we cannot find strong evidence and therefore relate all fluxes to the riverine export at the outlet. However, later in the text we discuss the possibility of groundwater export in the lowland-part of the catchment bypassing the gauging station (P27 L 23ff).

9. P21 L23: Export via groundwater can be a third explanation for the missing N.

Yes, you are right. We will specify that we assume the discussed aspects as potential, main reasons as we have no strong evidence for regional groundwater outflow out of the catchment (see comment above).

 I still don't understand the way that you've visualized this conceptual model. It's not clear what sections of the subsurface contribute which water during HFS and LFS. Why don't you use or modify the visualization by Rozemeijer & Broers, 2008?

Thank you for your suggestion. We checked the proposed figure of Rozemeijer & Broers (2008) that shows well separated activation of surface flow paths to a stream with increasing water level/ wetness in a lowland setting, but struggle to transfer this to our needs- showing the seasonally varying contribution over depth within one stream. Therefore, we would like to keep the figure without further changes. However, we carefully adjusted the figures caption to better describe the idea of the conceptual figure.

[revised manuscript text omitted]